EMBO
Molecular Medicine

# The enhancement of activity rescues the establishment of *Mecp2* null neuronal phenotypes

Linda Scaramuzza[1],[§] , Giuseppina De Rocco[1],[2],[†] , Genni Desiato[3],[4],[†] , Clementina Cobolli Gigli[1],[¶] , Martina Chiacchiaretta[5],[‡‡] , Filippo Mirabella[3],[6] , Davide Pozzi[3],[6] , Marco De Simone[7],[††] , Paola Conforti[7],[8] , Massimiliano Pagani[2],[7] , Fabio Benfenati[5],[9] , Fabrizia Cesca[5],[10] , Francesco Bedogni[1],[*],[‖],[‡] & Nicoletta Landsberger[1],[2],[**],[‡] [§§]

## Abstract

*MECP2* mutations cause Rett syndrome (RTT), a severe and progressive neurodevelopmental disorder mainly affecting females. Although RTT patients exhibit delayed onset of symptoms, several evidences demonstrate that MeCP2 deficiency alters early development of the brain. Indeed, during early maturation, *Mecp2* null cortical neurons display widespread transcriptional changes, reduced activity, and defective morphology. It has been proposed that during brain development these elements are linked in a feed-forward cycle where neuronal activity drives transcriptional and morphological changes that further increase network maturity. We hypothesized that the enhancement of neuronal activity during early maturation might prevent the onset of RTT-typical molecular and cellular phenotypes. Accordingly, we show that the enhancement of excitability, obtained by adding to neuronal cultures Ampakine CX546, rescues transcription of several genes, neuronal morphology, and responsiveness to *stimuli*. Greater effects are achieved in response to earlier treatments. *In vivo*, short and early administration of CX546 to *Mecp2* null mice prolongs lifespan, delays the disease progression, and rescues motor abilities and spatial memory, thus confirming the value for RTT of an early restoration of neuronal activity.

**Keywords** Ampakine; Mecp2; neuronal activity; neuronal maturation; Rett syndrome

**Subject Categories** Genetics, Gene Therapy & Genetic Disease; Neuroscience

## Introduction

Mutations in the X-linked methyl-CpG-binding protein 2 (*MECP2*) gene are associated with a number of neurological conditions among which the most frequent is Rett syndrome (RTT; Amir *et al*, 1999). RTT is the most common cause of severe intellectual disability in females, as it affects one female every 10,000 born alive (Chahrour & Zoghbi, 2007). Given the high levels of MeCP2 in the brain, the neurological features of RTT are by far the most thoroughly described. Small brain size and reduced weight, thin corpus callosum, and diminished cortical thickness are typical of the pathology (Armstrong *et al*, 2001; Carter *et al*, 2008; Belichenko *et al*, 2009). Hemizygous *Mecp2* null male mice recapitulate much of these defects in adult stages, therefore proving their face validity. However, diversely from most RTT girls, they feature significantly reduced life span. Defective neuronal features have also been reported in the RTT

1   Division of Neuroscience, IRCCS San Raffaele Scientific Institute, Milan, Italy
2   Department of Medical Biotechnology and Translational Medicine, University of Milan, Milan, Italy
3   IRCCS Humanitas Research Hospital, Milan, Italy
4   CNR Institute of Neuroscience, Milan, Italy
5   Center for Synaptic Neuroscience and Technology, Istituto Italiano di Tecnologia, Genova, Italy
6   Department of Biomedical Sciences, Humanitas University, Milan, Italy
7   Istituto Nazionale Genetica Molecolare "Romeo ed Enrica Invernizzi", Milan, Italy
8   Department of Biosciences, University of Milan, Milan, Italy
9   IRCCS Ospedale Policlinico San Martino, Genova, Italy
10  Department of Life Sciences, University of Trieste, Trieste, Italy
    *Corresponding author. Tel: +44 0 29208 76807; E-mail: bedognif@cardiff.ac.uk
    **Corresponding author. Tel: +39 02 26436278; E-mail: nicoletta.landsberger@unimi.it
    †These authors contributed equally to this work as second, third authors
    ‡These authors contributed equally to this work as senior authors
    §Present address: Department of Bioscience, University of Milan, Milan, Italy; Istituto Nazionale Genetica Molecolare "Romeo ed Enrica Invernizzi", Milan, Italy
    ¶Present address: Francis Crick Institute, London, UK
    ‡‡Present address: Department of Neuroscience, Tufts University School of Medicine, Boston, MA, USA
    ††Present address: Department of Radiation Oncology, Cedars-Sinai Medical Center, Los Angeles, CA, USA
    ‖Present address: Neuroscience and Mental Health Research Institute (NMHRI), Division of Neuroscience, School of Biosciences, Cardiff, UK
    §§Correction added on 9 April 2021, after first online publication: The affiliations and present addresses were corrected.

mouse models, including reduced soma size, dendritic branching, number of spines, and synaptic contacts (Guy *et al*, 2001; Kishi & Macklis, 2004; Fukuda *et al*, 2005; Chao *et al*, 2007; Belichenko *et al*, 2009; Baj *et al*, 2014; Rietveld *et al*, 2015; Bedogni *et al*, 2016; Sampathkumar *et al*, 2016). Importantly, although these phenotypes vary depending on the cell type, the age and type of *Mecp2* mutations, soma size is consistently reduced throughout development of mouse models of Rett syndrome, therefore appearing as a robust and reliable biomarker (Wang *et al*, 2013).

Besides morphological alterations, impaired neuronal functions were also observed in adult mice, resulting in a complex derangement of brain activity (Nelson & Valakh, 2015). *Mecp2* null cortical neurons feature reduced activity caused by both a selective impairment in excitatory transmission and a reduced connectivity between excitatory neurons (Dani *et al*, 2005; Dani & Nelson, 2009; Shepherd & Katz, 2011; Sceniak *et al*, 2016). *Mecp2* KO visual cortex manifest similar deficiency in neuronal and network activity that, however, appears to result from stronger inhibition (Durand *et al*, 2012). On the contrary, the adult and symptomatic *Mecp2* null hippocampus (but not the presymptomatic one) suffers from elevated neuronal activity and occluded LTP caused by potentiated synapses (Li *et al*, 2017).

The outbreak of symptoms typically follows an apparently normal developmental phase that lasts 6–18 months in humans (Cosentino *et al*, 2019) and roughly 40 days in *Mecp2* null animals (Guy *et al*, 2001; Chen *et al*, 2001). However, subtle symptoms in RTT girls have been described even before the onset of the overt phase of the pathology (Fehr *et al*, 2011; Dolce *et al*, 2013; Marschik *et al*, 2013; Cosentino *et al*, 2019). Similarly, animal models display defects already during embryonic development (Bedogni *et al*, 2016; Cobolli Gigli *et al*, 2018; Mellios *et al*, 2018) or immediately after birth (Dani *et al*, 2005; Picker *et al*, 2006; Chao *et al*, 2007). Such evidence proves that the lack of MeCP2 affects early development of the mammalian brain although the phenotypic consequences of it become overt later on (Ip *et al*, 2018; Cosentino *et al*, 2019).

In line with the role of Mecp2 as master regulator of transcription (Skene *et al*, 2010; Lagger *et al*, 2017; Shah & Bird, 2017), gene expression is affected in null samples already during early development both *in vivo* (Cobolli Gigli *et al*, 2018) and *in vitro* (Mellios *et al*, 2018), as well as neuronal morphology and responsiveness to external *stimuli* (Bedogni *et al*, 2016). We thus hypothesized that these early developmental defects could be a key element to explain the thoroughly described poor maturity of adult *Mecp2*-deficient neuronal networks. Indeed, while neuronal activity is surely involved in the maintenance of mature synaptic structures (Greer & Greenberg, 2008; Flavell & Greenberg, 2008), early patterns of spontaneous activity have been reported already during embryonic and early postnatal cortical development (Corlew *et al*, 2004; Platel *et al*, 2005; Spitzer, 2006). However, such spontaneous activity cannot be simply considered a measure of the gradually increasing maturation of membrane excitability, as it also drives neurodevelopmental processes that include refinement of neuronal identity and newborn neurons migration, survival, and connectivity (Weissman *et al*, 2004; Bonetti & Surace, 2010; Yamamoto & Lopez-Bendito, 2012; Luhmann *et al*, 2015). Indeed, it has been proposed that neuronal activity is required at all developmental stages, as it refines signaling via gene expression, providing checkpoints to validate or modulate genetic programs essential for the establishment of proper neuronal maturity (Spitzer, 2006). We thus tested the possible causative link

between poor maturity and reduced neuronal activity in null samples by pharmacologically stimulating null neurons within early time windows. To focus on the earliest steps of neuronal maturation, we used neurons differentiated from cortical neuroprecursors (NPCs), which recapitulate the *in vivo* process of neuronal differentiation and the subsequent steps of gliogenesis (Paridaen & Huttner, 2014). Functional *in vitro* assessments were produced using differentiated neurons or primary neuronal cultures. Eventually, we validated our results *in vivo* using *Mecp2* null animals.

Our *in vitro* data demonstrate that the enhancement of activity rescues *Mecp2* null neurons morphology and expression of selected transcripts, thus restoring impaired network functions. Accordingly, we show that the exposure of *Mecp2* null animals to Ampakine CX546 within an early and short postnatal time window prolonged life span and ameliorated their behavioral scoring. Importantly, both *in vivo* and *in vitro* rescue effects lasted long after the exposure to the drug, implying the long persistence of such effects. Our results suggest that the enhancement of neuronal activity in *Mecp2* null neurons sets in place plasticity mechanisms that mitigate maturation defects likely through the modulation of a feed-forward cycle (Spitzer, 2006). We suggest that the time frame in which such mechanisms occur highlights an important early molecular phase of the pathology, during which null neurons are affected by a number of transcriptional impairments that are prodromal of the subsequent onset of overt symptoms. Therapeutic strategies acting within this early molecular phase might be more effective in readdressing the pathological trajectory of null neurons development toward more physiological directions than later treatments.

## Results

### Lack of Mecp2 does not affect the early commitment of cultured cortical NPCs

To focus on early mechanisms of maturation of *Mecp2* null neuronal networks, we collected neuroprecursor cells (NPCs) from E15 cerebral cortex and induced their *in vitro* differentiation as previously described (Magri *et al*, 2011; Cobolli Gigli *et al*, 2018). Being synchronously generated, the stage of differentiation of such populations is roughly comparable. Defects masked in heterogeneous populations are consequently easier to detect, an advantage in the case of Mecp2 studies, where in most cases defects are very subtle (Bedogni *et al*, 2014). Moreover, this model enables to reduce the use of mice and is suitable for pharmacological manipulations (Gorba & Conti, 2013).

NPCs spontaneously proliferate in a self-renewing manner in the presence of the mitogens EGF and FGF2 (Gritti *et al*, 1999), while the presence of fetal bovine serum drives their differentiation into the three main cellular populations of the brain: neurons, astrocytes, and oligodendrocytes (Magri *et al*, 2011; Fig 1A). We measured the distinct cellular populations at different developmental stages of NPCs differentiation by immunostaining wild-type (wt) and *Mecp2* null cultures for Tuj1, Gfap, and Olig2. We found that the lack of Mecp2 did not affect the percentage of neurons, astrocytes, and oligodendrocytes at any differentiation stage (Fig 1B–J), in line with previous observations (Kishi & Macklis, 2004). In particular, neurons contributed for a percentage of roughly $27.2 \pm 1.1$ and $22.2 \pm 0.6$ at

DIV8 and $21.6 \pm 1.08$ and $19.7 \pm 1.6$ at DIV22 in wt and null samples, respectively (two-way ANOVA followed by Bonferroni *post hoc* test; Fig 1D). The percentage of Gfap positive cells increased from $35.7 \pm 0.8$ and $39.4 \pm 1.06$ at DIV8 to $69.8 \pm 4.7$ and $69.4 \pm 4.4$ at DIV22 in wt and null samples, respectively (Fig 1G). Gfap-positive cells were, therefore, generated at a later time point than neurons, in line with the *in vivo* dynamics of glial cells genesis and differentiation (Paridaen & Huttner, 2014). The percentage of oligodendrocytes (roughly 7% in both genotypes) remained equal throughout differentiation (two-way ANOVA followed by Bonferroni

*post hoc* test; Fig 1J). At DIV8, we found that although some cells expressed Map2, a typical marker of mature neurons (Fig EV1A), roughly 85% of wt and null cells were positive for Nestin (Fig EV1C), thus suggesting that immature NPCs were still present at that stage. Such population eventually disappeared at DIV22 (Fig EV1B), in line with the decreasing expression of the cell cycle marker Ki67 over time (*$P < 0.05$, **$P < 0.01$, ****$P < 0.0001$, two-way ANOVA followed by Bonferroni *post hoc* test; Fig EV1D). We conclude that at DIV22 most cells terminally differentiated into their specific fate with no influence of the genotype on final commitment.

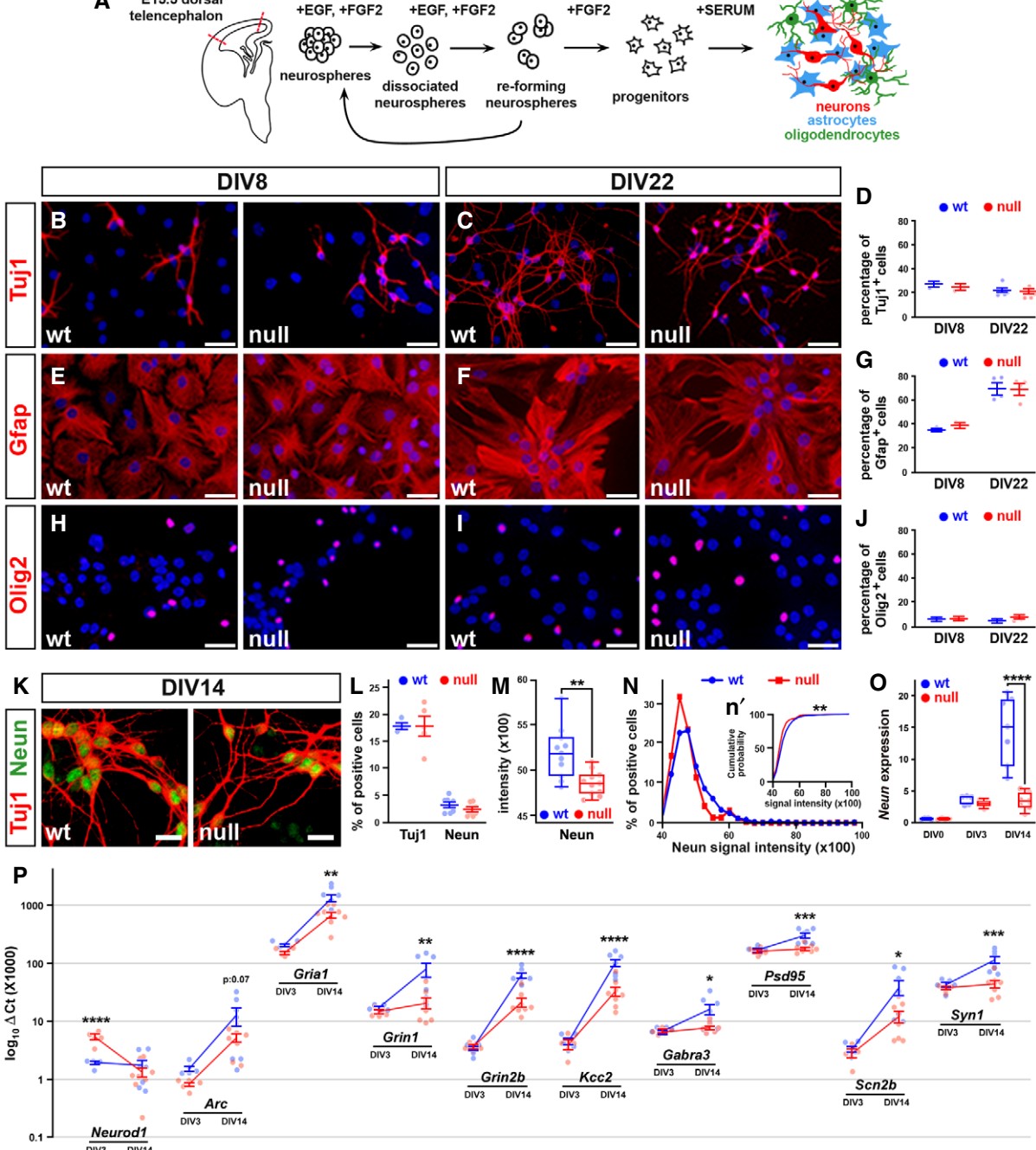

**Figure 1.**

**Figure 1. Lack of Mecp2 does not affect the commitment of NPCs but delays their differentiation.**

A  Schematic representation of the differentiation protocol and the cell types composing cultures differentiated from NPCs (adapted from Carlessi et al, 2013).

B–J  Representative immunofluorescences of the cell types composing the wt and *Mecp2* null cultures differentiated from NPCs. Antibodies against Tuj1, Gfap, and Olig2 were used to recognize neurons, astrocytes, and oligodendrocytes. In D, G, and J, the percentage of cells positive (mean ± SEM) for each marker was counted over the total of DAPI-positive cells (100%). No difference between genotypes was highlighted (two-way ANOVA); the variation through time resulted significant for the levels of Tuj1 and Gfap (two-way ANOVA: $F_{(1, 16)}$ = 9.542, P-value < 0.05 and $F_{(1, 10)}$ = 68.95, P-value < 0.0001, respectively). Sample size: $n \geq 3$ wells obtained from two independent preparations; for each well, 10 random fields were acquired. Scale bars: 30 μm.

K  Representative immunostaining of Neun (green) and Tuj1 (red)-positive neurons at DIV14. Scale bars: 15 μm.

L, M  Graphs show the percentage of Neun and Tuj1-positive cells and the average Neun intensity in wt and null samples (L: mean ± SEM; M: central band shows the median, boxes show the lower and upper quartiles, whiskers show max and min values). The percentage of cells positive for each marker was counted over the total of DAPI-positive cells. Student's *t*-test highlighted a significant decrease of Neun intensity in null samples. **P-value < 0.01. Sample size: $n \geq 10$ wells obtained from two independent preparations.

N, n′  The binned distribution of Neun intensity is plotted as both frequency distribution graphs, in panel N, and cumulative probability plot in panel n′. The null neurons distribution is significantly shifted toward low levels of Neun intensity, Kolmogorov–Smirnov test **P-value < 0.01. Each value in panel N is represented as mean ± SEM obtained by measuring 900 cells from $n \geq 10$ wells derived from two different preparations.

O  *Neun* expression levels assessed at DIV0, DIV3, and DIV14 are represented as $2^{-\Delta ct}$. Box plots show the median (central band), the lower and upper quartiles (boxes) and the max and min values (whiskers) from at least 5 samples obtained from two independent preparations. Two-way ANOVA indicated both a significant genotype effect ($F_{(2, 28)}$ = 19.60; P < 0.0001) and time effect ($F_{(2, 28)}$ = 93.70; P < 0.0001). Bonferroni *post hoc* test revealed significant differences between genotypes at DIV 14 but not at DIV0 and DIV 3. ****P-value < 0.0001.

P  Scatter plots describing the transcriptional levels of selected differentially expressed genes in wt and *Mecp2* null cultures differentiated from NPCs and analyzed at DIV3 and DIV14. For each gene, two-way ANOVA and Šidák multiple comparison test were used to assess difference between genotypes at the two selected time points (*P < 0.05, **P < 0.01, ***P < 0.001, ****P < 0.0001). Values are represented as mean ± SEM. Sample size: ≥5 samples obtained from two independent preparations.

Source data are available online for this figure.

To further support these findings, we repeated the analysis on NPCs derived from heterozygous $Mecp2^{-/+}$ female cerebral cortices, which are generally subjected to random X chromosome inactivation (Fig EV1E–H). Consistent with our previous results (Fig 1D and G), at DIV22 we found no different percentage of either neurons (Tuj1⁺; 47.8 ± 1.4 and 52.2 ± 1.4) or glial cells (Gfap⁺; 48.9 ± 1.5 and 51.1 ± 1.5) between the Mecp2-positive and Mecp2-negative populations (Student's *t*-test; Fig EV1F–H).

**Lack of Mecp2 affects transcription of *in vitro* differentiated NPCs**

To verify whether our model recapitulated the typical delay in neuronal maturation (Bedogni et al, 2016; Mellios et al, 2018; Cobolli Gigli et al, 2018), we exploited the expression levels of Neun, a neuronal-specific marker that increases with maturation at a slower rate compared to Tuj1 (Menezes & Luskin, 1994; Walker et al, 2007). We analyzed null and wt cultures at DIV14 (Fig 1K–O) and found that the number of Neun and Tuj1 positive cells overlapped between wt and null samples (Neun: 3.2 ± 0.4 and 2.5 ± 0.3 in wt and null respectively; Tuj1: 17.9 ± 0.4 and 17.8 ± 1.8 in wt and null, respectively; two-way ANOVA followed by Bonferroni *post hoc* test; Fig 1L). However, the intensity of the Neun signal was significantly decreased in null samples (5190 ± 85.5 in wt and 4855 ± 39.98 in null samples; **P < 0.01, Student's *t*-test; Fig 1M), as well as the percentage of null cells expressing low levels of it (**P < 0.01, Kolmogorov–Smirnov test; Fig 1N and n′).

Moreover, in wt cultures the mRNA levels of Neun increased with time, while the expression of Neun did not change in *Mecp2* null cultures between DIV3 and DIV14 (****P < 0.0001, two-way ANOVA followed by Bonferroni *post hoc* test; Figs 1O and EV2). To corroborate the existence of transcriptional differences among the two experimental groups, we measured the expression of two sets of genes that included either 16 mRNAs typically expressed by NPCs or 51 transcripts mainly expressed by post-mitotic neurons (Appendix Table S1). In line with previous studies (Cobolli Gigli

et al, 2018), at the earliest time point (DIV3), changes were mainly observed in transcripts selected for their expression in mitotic or early post-mitotic neurons, such as the increased expression of Neurod1 (Fig 1P). At DIV14 we found reduced levels of transcripts involved in the responsiveness to external *stimuli* (such as ionic channel subunits, Glutamate and GABA receptors) and in broader neuronal functions (Syn1, Psd95) (*P < 0.05, **P < 0.01, ***P < 0.001, ****P < 0.0001; two-way ANOVA followed by Šidák multiple comparison test; Fig 1P). These results were recapitulated by principal component analyses (PCA) used to qualitatively visualize the segregation of the two genotypes based on the differential expression of markers of either progenitor or mature post-mitotic cells. In fact, by applying a set of genes defining immature neurons we unmasked at DIV0 a segregation (Fig EV2A–A′) that disappeared at later time points. Conversely, the use of markers of post-mitotic neurons showed a more evident separation at DIV14 (Fig EV2B–B′). Notably, the deregulation of several genes was also evident at DIV21 (*P < 0.05, **P < 0.01, ***P < 0.001; two-way ANOVA followed by Bonferroni *post hoc* test; Fig EV2C), thus confirming that NPC-derived *Mecp2* null neurons well recapitulate the early onset and persistency of transcriptional changes due to the lack of Mecp2 (Bedogni et al, 2016; Cobolli Gigli et al, 2018).

**The *in vitro* established *Mecp2* null neuronal networks are functionally and morphologically immature**

To verify the impact of maturation defects on the activity of *Mecp2* null neuronal networks, we next focused on functional assessments. We plated dissociated NPCs at high density (8,000 cells/μl) on multi-electrode array (MEA) chips to record spontaneous electrical activity (Fig 2A and B). The same population of cells was recorded at two time points (DIV18 and 22) to monitor the functional development of cell-to-cell communication and the generation of an active network over time.

The spontaneous activity of the developing networks was assessed through five parameters: number of active electrodes,

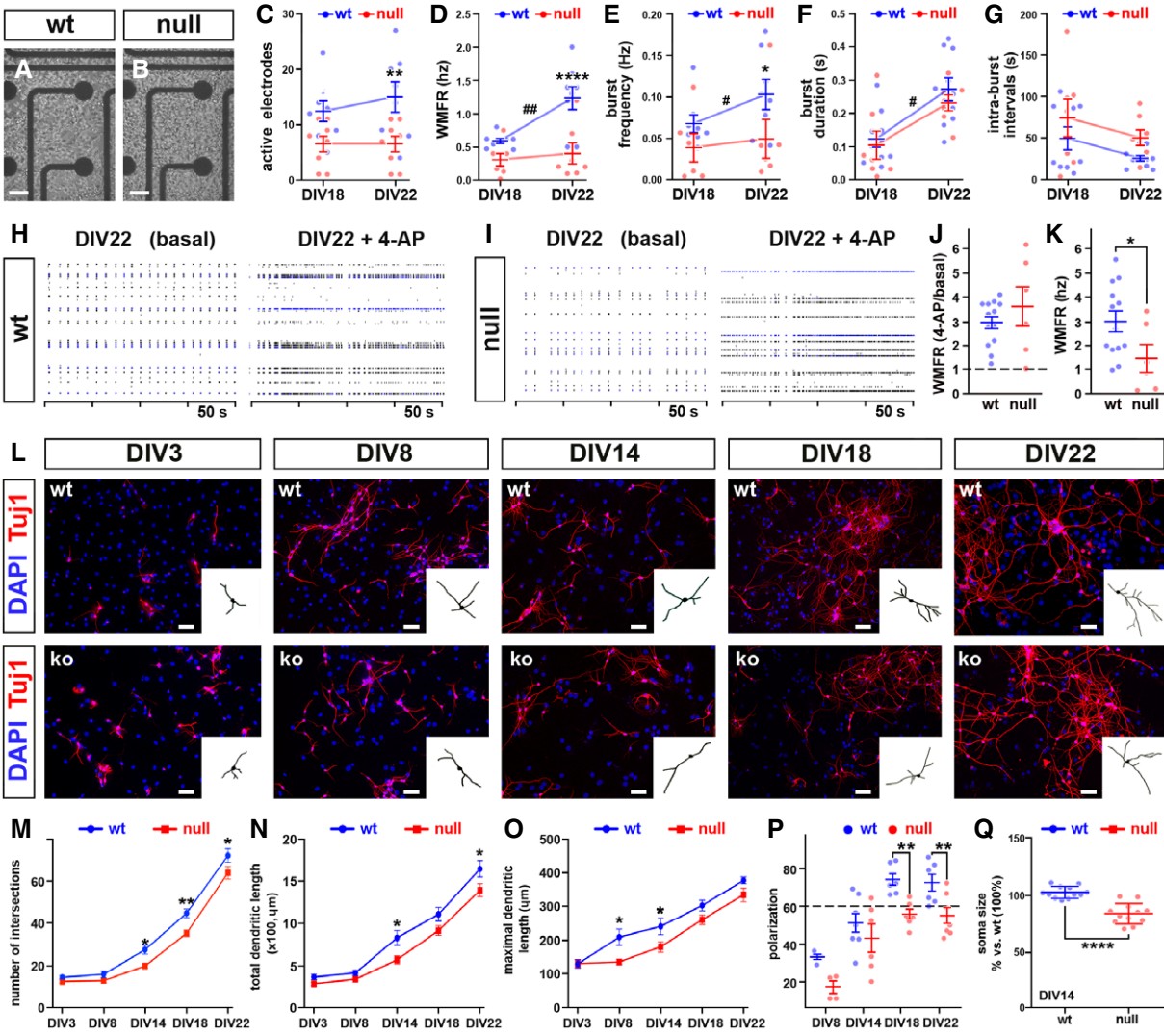

**Figure 2.   Neurons derived from *Mecp2* null NPCs are morphologically more immature than wt controls and establish less active networks.**

A, B     Wt or *Mecp2* null NPCs plated on the grid of 64 MEA electrodes. Scale bars: 50 μm.

C–G     Line graphs represent the changes through time of 5 parameters describing neuronal activity. Difference between genotypes and through time were analyzed using two-way ANOVA for repeated measures and Bonferroni *post hoc* test; * refers to differences between genotypes (*P-value < 0.05; **P-value < 0.01; ****P-value < 0.0001); # refers to differences between the two time points (#P-value < 0.05; ##P-value < 0.01). Values are represented as mean ± SEM. Sample size: n = 8 wells deriving from three independent preparations. Values are represented as mean ± SEM.

H–K     Raster plots (H and I) and their quantitation under basal conditions and upon 100 μM 4-AP exposure. The magnitude of the response after stimulation is expressed either as fold change (J) or as the averaged maximal values reached after 4-AP exposure (K). Dotted line in J represents untreated controls (set to 1). Values are represented as mean ± SEM. Student's *t*-test, *P-value < 0.05. Sample size n = 8 wells deriving from three independent preparations.

L        Wt and null neurons counterstained with an antibody against Tuj1 and DAPI at different time points. The binary mask is representative of a neuron used to study morphology. Scale bars: 30 μm.

M–Q     Each graph represents a different morphological parameter analyzed at selected time point. Two-way ANOVA indicated both a significant time effect (F 4, 20) = 282.5 and P < 0.0001 for the number of intersections; (F 4, 26) = 104.5 and P < 0.0001 for the total dendritic length; (F 4, 20) = 58.51 and P < 0.0001 for the maximal dendritic length; (F 3, 22) = 38.32 and P < 0.0001 for polarization) and genotype effect ((F 1, 20) = 25.1 and P < 0.0001 for the number of intersections; (F 4, 26) = 16.0 and P = 0.005 for the total dendritic length; (F 1, 20) = 17.24 and P = 0.005 for the maximal dendritic length; (F 1, 22) = 18.76 and P < 0.003 for polarization). Bonferroni multiple comparison test was used to assess difference between wt and null neurons at each analyzed time point. The dotted line in panel P represents the cut-off above which a culture is defined polarized (Horton *et al*, 2006). Statistical significance for Q was assessed by Student's *t*-test. Values are represented as mean ± SEM; *P-value < 0.05; **P-value < 0.01; ****P-value < 0.0001. Sample size: n ≥ 3 wells deriving from three independent preparations; for each well, we measured at least 15 cells.

Weighted Mean Firing Rate (WMFR), frequency and duration of spontaneous bursts, and intra-burst intervals (Fig 2C–G). The maturation of wt network activity was demonstrated by the significant increase of WMFR, burst frequency, and burst duration at DIV22 compared to DIV18 and the increasing trend in the number of active electrodes in the network (##P < 0.01, two-way ANOVA followed by

Bonferroni *post hoc* test), in line with published observations on the same model (de Groot *et al*, 2014). The lack of Mecp2 instead drove a markedly alteration of this pattern, with null cultures showing lack of functional maturation between DIV18 and DIV22, as indicated by the analysis of active electrodes, WMFR, burst frequency, and duration (two-way ANOVA followed by Bonferroni *post hoc* test; Fig 2C–F). The intra-burst intervals showed a similar trend between the two genotypes (Fig 2G). These data suggest that, differently from wt, null neurons did not integrate into an increasingly cohesive network, indicative of a defect in neuronal maturation. Noticeably, the parameters describing network activity in our model were lower in magnitude compared to primary neurons, as expected given the different level of maturity displayed by the two *in vitro* models (Chiacchiaretta *et al*, 2017).

After measuring spontaneous activity, we studied the evoked response upon exposure to 100 μM of the voltage-activated $K^+$ channels blocker 4-aminopiridine (4AP) at DIV22 (Fig 2H–K). Although the fold increase in WMFR was comparable between the two genotypes ($2.93 \pm 0.3$ in wt and $3.6 \pm 0.8$ ko samples; Student's *t*-test; Fig 2J), when considering the absolute WMFR values, the response was twofold higher in wt samples compared to null samples ($2.99$ Hz $\pm 0.4$ in wt and $1.52$ Hz $\pm 0.5$ ko samples; $*P < 0.05$; Student's *t*-test; Fig 2K). These results demonstrate that networks established from Mecp2 null differentiated NPCs are less active compared to wt, in line with previous data (Kron *et al*, 2012; Bedogni *et al*, 2016; Katz *et al*, 2016; Sceniak *et al*, 2016).

We next analyzed the growth of dendritic branches. The dendritic complexity analyzed in wt and Mecp2 null neurons increased through time in both genotypes (Fig 2L). However, starting from DIV14, null neurons exhibited a poorer dendritic growth compared to wt, as we detected a significant reduction of both number of intersections and total dendritic length ($**P < 0.01$, $*P < 0.05$; two-way ANOVA followed by Bonferroni *post hoc* test; Fig 2M and N), while maximal dendritic length appeared reduced at earlier time points ($**P < 0.01$, $*P < 0.05$; two-way ANOVA followed by Bonferroni *post hoc* test; Fig 2O). Similarly, the percentage of polarized null neurons (see Materials and Methods for definition) resulted significantly diminished from DIV18 ($73.8\% \pm 3.2$ and $55.8\% \pm 2.7$ in wt and null neurons, respectively; $*P < 0.05$; two-way ANOVA followed by Bonferroni *post hoc* test; Fig 2P). Eventually, and in line with previous findings (Wang *et al*, 2013; Bittolo *et al*, 2016; Sampathkumar *et al*, 2016), NPC-derived null neurons at DIV14 showed a reduction in soma size ($85.8$ μm$^2$ $\pm 1.8$ and $61.7$ μm$^2$ $\pm 0.9$ in wt and null neurons, respectively; $***P < 0.001$, Student's *t*-test; Fig 2Q).

Overall, these functional and morphological data confirm that our model reproduces many of the typical maturation defects generated *in vivo* and *in vitro* by the absence of Mecp2 (Baj *et al*, 2014; Rietveld *et al*, 2015; Bedogni *et al*, 2016; Sampathkumar *et al*, 2016).

### The enhancement of glutamatergic transmission within an early time window produces rescue effects that persist at later time points

Our hypothesis is that reduced activity-dependent signaling in Mecp2 null neurons contributes to their defective transcription and morphological immaturity, which further reinforce a defective response to *stimuli*. We thus studied whether a pharmacological enhancement of

neuronal excitability could rescue the observed maturation defects. With this aim, maturing neurons were treated with Ampakine CX546, a positive modulator of glutamatergic transmission. CX546 binds the glutamate-activated AMPA receptor on the subunits GRIA1 and GRIA2 reducing desensitization and slowing channel closure (Nagarajan *et al*, 2001; Lynch & Gall, 2006). Initially, we measured the levels of *Gria2* at different developmental stages to verify whether both wt and null cultures had comparable potential to respond to the drug. We found that wt and *Mecp2* null neurons expressed similar amounts of the AMPA subunit at DIV3 and 7, while at DIV14 null neurons showed a significantly reduced expression compared to wt ($0.60 \pm 0.03$ in wt and $0.46 \pm 0.03$ in null samples; $**P < 0.01$, two-way ANOVA followed by Bonferroni *post hoc* test; Fig EV3B). Moreover, we investigated the involvement of the AMPA receptor in mediating Ampakine effects by acutely exposing cells to CX546 alone or simultaneously with the AMPA receptor antagonist NBQX. As expected, NBQX inhibited the drug effects on Akt phosphorylation ($*P < 0.05$; one-way ANOVA followed by Tukey's *post hoc* test; Fig EV3A). Based on published data, we then tested longer exposure of two different doses of CX546 (10 and 20 μM; Schitine *et al*, 2012). Daily addition of 10 μM CX546 from DIV7 to DIV10 did not produce toxic effects, while chronic exposure to 20 μM (from DIV1 to DIV10) resulted in only 50% cell survival (Fig EV3C). We thus designed a protocol in which 20 μM CX546 was added every day to the cultures for 4 days and then washed out. To assess whether different time windows of CX546 exposure could produce different rescue effects, CX546 was added either from DIV3 to DIV6 or from DIV7 to DIV10 (Fig 3A) followed by washout. Samples were then collected for morphological and transcriptional assessments at DIV14. As shown in Fig 3B and C, CX546 treatments produced a significant amelioration of several morphological parameters in null neurons, including dendritic length, number of neurite intersections, and soma size. Notably, the effects were slightly stronger when the drug was early administered ($**P < 0.01$, $*P < 0.05$; one-way ANOVA followed by Tukey's *post hoc* test); no trophic effect was observed in wt samples (Appendix Fig S1A–D). To confirm the contribution of reduced activity to the typical immaturity displayed by *Mecp2* null neurons, we also tested the effects of the addition of 4 mM KCl on neuronal morphology. In line with our previous results, we found that a chronic exposure of differentiating neuronal cultures to KCl rescued the dendritic complexity of *Mecp2* null neurons ($***P < 0.001$, $**P < 0.01$, $*P < 0.05$; one-way ANOVA followed by Tukey's *post hoc* test; Fig EV4A–D). Importantly, such treatment resulted more effective when applied early during neuronal maturation compared to a late treatment (from DIV0 to DIV7 followed by washout or from DIV7 to DIV14, respectively; $***P < 0.001$, $**P < 0.01$, $*P < 0.05$; two-way ANOVA followed by Bonferroni *post hoc* test; Fig EV4E–G). Next, we assessed transcription on RNA samples prepared from wt controls and null neurons treated with either Ampakine or vehicle and collected at DIV14 (Fig 3D–H). Initially, as for Fig EV2A and B, we used PCA to qualitatively address whether CX546 reduced the transcriptional distance of null from wt samples. To this purpose, we analyzed only those transcripts that resulted unaffected by batch effects (43 genes; Appendix Fig S2). As depicted in panel E, when the analysis was performed after early CX546 exposure (DIV3-DIV6), the transcriptional distance from wt and null treated samples appeared reduced compared to basal condition (Fig 3D). In line with this, a detailed

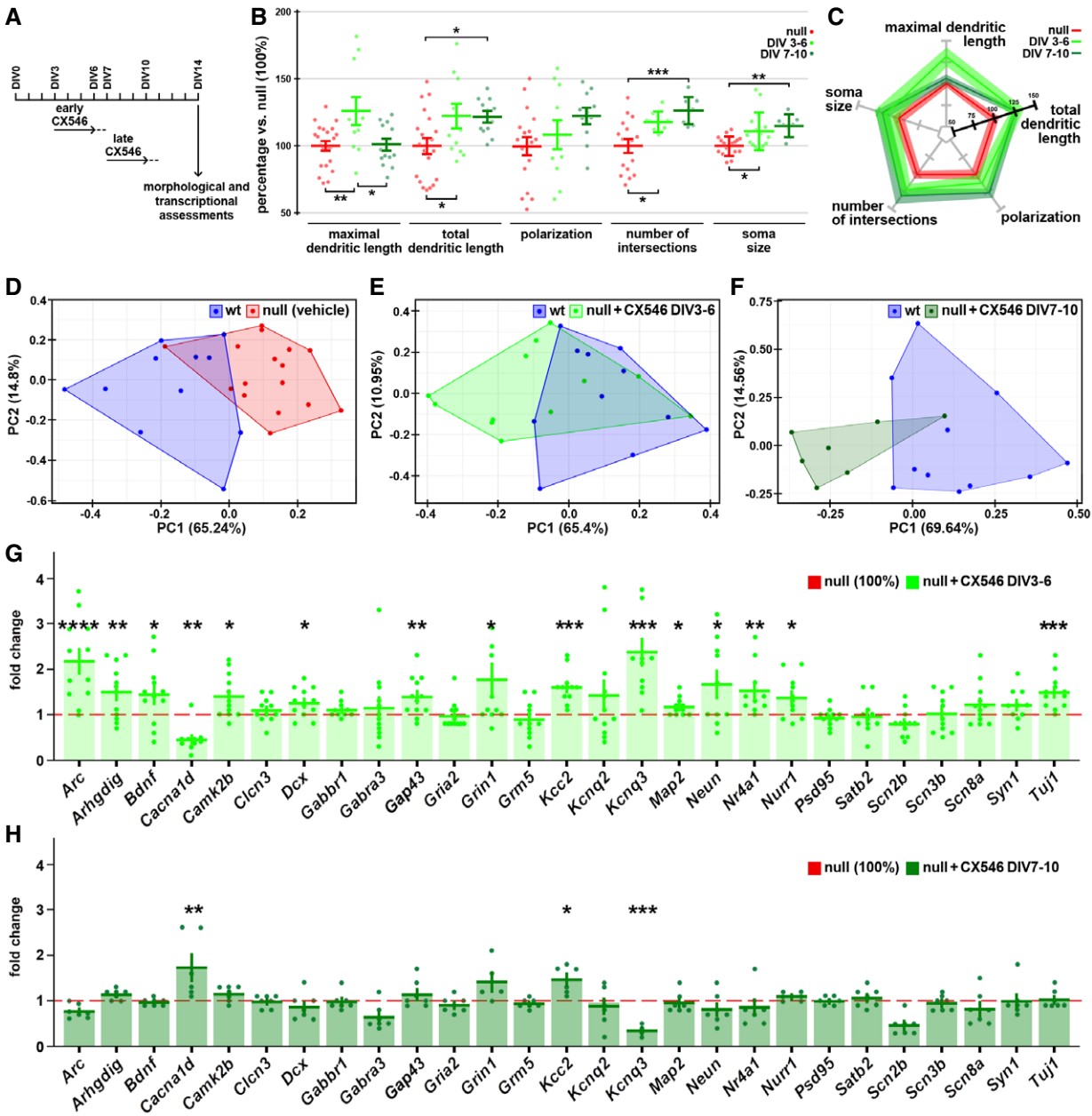

**Figure 3. Early exposure to CX546 rescues selected morphological and transcriptional defects of null neurons.**

A       Schematic representation of the CX546 treatment. The drug was added to the cell medium from DIV3 to DIV6 (early) and from DIV7 to DIV10 (late); a complete washout of the medium was performed at the end of each treatment (DIV7 and DIV11, respectively).

B, C    Five morphological parameters were measured in untreated and treated null neurons at DIV14. One-way ANOVA indicated a significant treatment effect for the maximal dendritic length ($F = 6.306$; $P = 0.0039$), total dendritic length ($F = 4.880$; $P = 0.0126$), number of intersections ($F = 10.70$; $P = 0.0003$), and soma area ($F = 7.721$; $P = 0.0015$). Tukey's multiple comparison test was used to assess differences between null neurons and early and late treated neurons. *$P$-value < 0.05; **$P$-value < 0.01; ***$P$-value < 0.001. The variation of each of the selected parameters is summarized in the radar plot (C). Sample size (panels B, C): $n \geq 18$ wells for ctrl group and $n \geq 10$ wells for the treatment groups deriving from three independent preparations; for each well, we measured at least 15 cells. Each point is representative of a single well. Values are represented as mean $\pm$ SEM.

D–H    The three PCA analyses are based on the 43 genes selected in Appendix Fig S2 and highlight the transcriptional differences between wt (in blue), ko (in red), and ko CX546 treated (light and dark green for early and late treatment, respectively) samples. Bar graphs represent the expression values (mean $\pm$ SEM) of 27 genes, selected for their down-regulation in null samples at DIV14 ($P$-value < 0.07; see Appendix Fig S2), after CX546 administration in the two different time windows (early in G and late in H). One-way ANOVA followed by Dunnett's multiple comparison test was used to compare the expression of each gene between *Mecp2* null early- and late-treated samples and their corresponding untreated controls (set to 1, red dotted line). *$P$-value < 0.05; **$P$-value < 0.01, ***$P$-value < 0.001, ****$P$-value < 0.0001. Sample size: $n \geq 8$ wells deriving from two independent preparations. Each point is representative of a single well.

representation of the 27 transcripts selected for their down-regulation between wt and null samples (Appendix Fig S2), showed that 14 of them were increased to levels that were significantly higher compared to untreated null samples (**$P < 0.01$, *$P < 0.05$; one-way ANOVA followed by Dunnett's *post hoc* test; red dotted line in Fig 3G). Importantly, CX546 appeared more effective when added from DIV3 to DIV6 compared to DIV7-DIV10, as shown by PCA analyses in Fig 3E and F. Accordingly, among the selected 27 genes only 2 were significantly increased (***$P < 0.001$, *$P < 0.05$; one-way ANOVA followed by Dunnett's *post hoc* test; Fig 3H). To be noticed, these treatments produced limited or no significant transcriptional change on wt neurons (Appendix Fig S1E and F).

We next tested the effectiveness of CX546 on primary neuronal cultures produced from null and wt E15 embryonic cortices, a model that has been thoroughly used to highlight functional defects driven by the lack of Mecp2. In line with our previous observations (Bedogni *et al*, 2016), primary neurons responsiveness to extracellular *stimuli* was impaired by the lack of Mecp2, as indicated by the analysis of intracellular $Ca^{2+}$ transients induced by 100 μM NMDA (Fig 4A–C). The two bell-shaped curves representing the distribution of the responses of DIV14 wt (in blue) and null samples (in red) in fact did not overlap, as the null curve shifted toward smaller responses. However, after early exposure to CX546, the null curve overlapped with that of wt (Fig 4B), therefore suggesting that the early treatment rescued the reduced responsiveness to NMDA. On the contrary, the late treatment with CX546 produced no rescue (Fig 4C), reinforcing the higher effectiveness of the early treatment (***$P < 0.001$, *$P < 0.05$; two-way ANOVA followed by Bonferroni *post hoc* test).

Next, we focused on the excitatory-to-inhibitory developmental switch of GABA responses. Such process depends on the

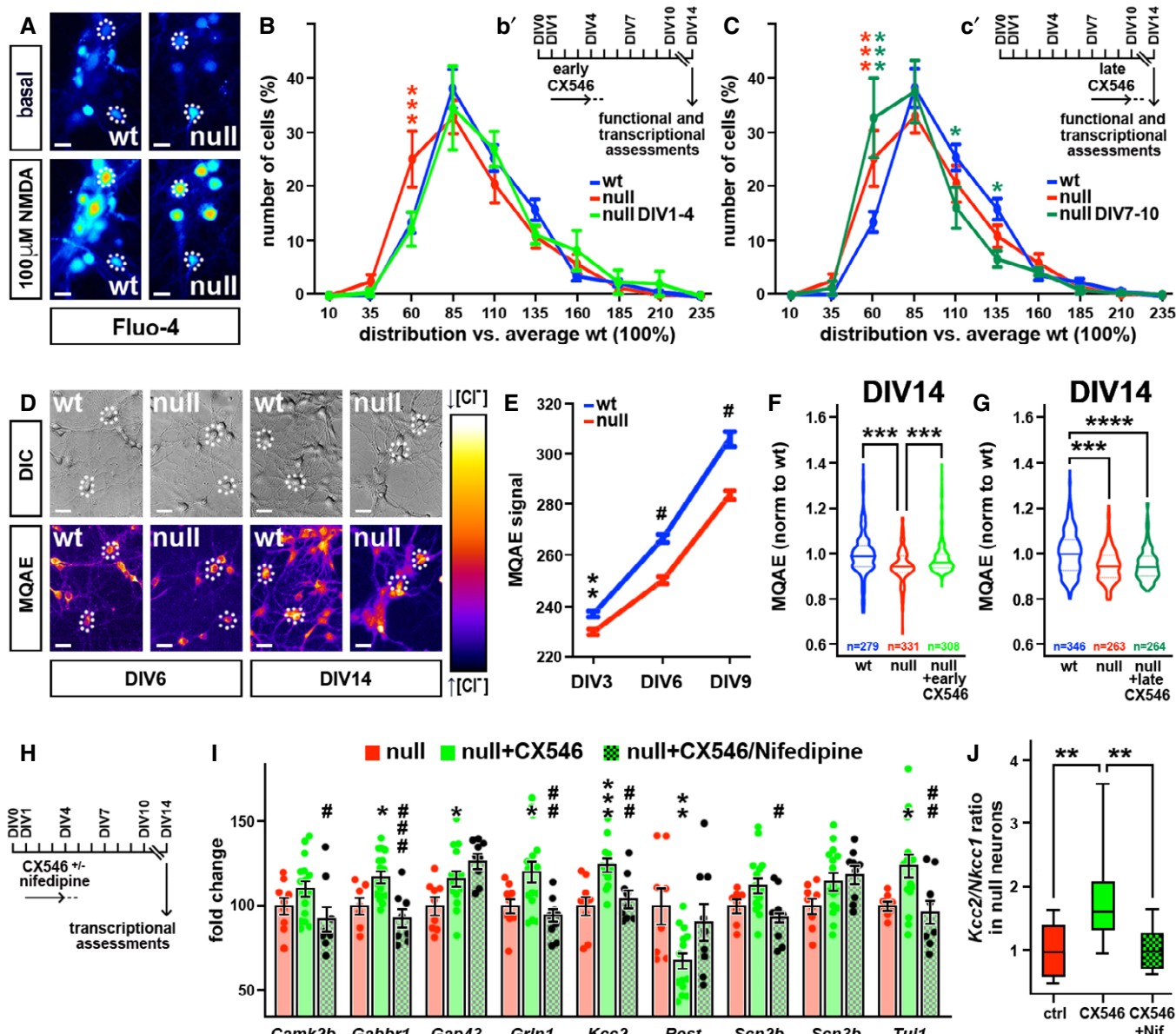

**Figure 4.**

◀

**Figure 4.** *Mecp2* **null neuronal network responsiveness is rescued by early treatment with CX546.**

A  Representative images of DIV14 wt and *Mecp2* null primary cortical neurons loaded with Fluo-4 before and after NMDA 100 μM exposure. High fluorescent signals correspond to high calcium levels. Dotted circles are representative of the Region Of Interest (ROI) used during the analysis. Scale bars: 20 μm.

B, C  The graphs represent the binned distribution of the magnitude of NMDA-induced calcium transients in wt (in blue), *Mecp2* null (in red), and treated *Mecp2* null (light and dark green for early and late treatment, respectively). The distribution of responses on X-axis are expressed as percentage compared to wt (100%), values on Y-axis refer to percentage of cells over total (100%). Two-way ANOVA followed by Bonferroni *post hoc* multiple comparison test was used to compare calcium responses between wt, *Mecp2* null and *Mecp2* null early (B) and late treated (C) neurons. *$P$-value < 0.05 ***$P$-value < 0.005. Red *: comparison between wt and ko; green *: comparison between wt and treated ko. Inserts b' and c' represent the protocols of CX546 exposure; dotted lines indicate the washout of the drug. Values are represented as mean ± SEM. Sample size: $n = 11$ and 10 animals for wt and *Mecp2* null control groups (for a total of 710 and 670 cells measured, respectively) and $n = 4$ and 6 animals for early and late CX546-treated groups (for a total of 270 and 360 cells measured, respectively).

D  Representative images of DIV6 and DIV14 wt and *Mecp2* null primary cortical neurons loaded with MQAE dye. High levels of fluorescence correspond to low level of intracellular chloride. Dotted circles are representative of the Region Of Interest (ROI) used during the analysis. Scale bars: 20 μm.

E  Line graphs represent intracellular chloride concentration at different consecutive time points in wt and *Mecp2* null neurons. Two-way ANOVA indicated a significant effect for both time ($F(2, 1541) = 840.9$ and $P < 0.0001$) and genotype ($F(1, 1541) = 155.7$ and $P < 0.0001$). Bonferroni *post hoc* test revealed a statistically significant reduction of *Mecp2* null MQAE intensity at all the analyzed time points. **$P$-value < 0.01; #$P < 0.0001$. Values of MQAE intensity are represented as mean ± SEM. Sample size: $n \geq 200$ cells from three different embryos per genotype.

F, G  Violin plots represent the average values of MQAE intensity for early (F) or late (G) treated null neurons and their relative controls. One-way ANOVA followed by Tukey's multiple comparison test was used to compare null control neurons and null-treated neurons to wt neurons. ***$P < 0.001$; ****$P < 0.0001$. The central line of the violin plot is the median while the two dotted lines represent the upper and lower quartiles. Sample size: $n \geq 300$ cells from five different embryos per genotype.

H  Schematic representation of CX546 and Nifedipine administration; dotted line indicates drug wash out.

I, J  In I, one-way ANOVA was used to compare the expression of 9 neuronal genes between *Mecp2* null primary neurons (set to 100), null neurons treated with CX546 from DIV1 to DIV4 and null neurons treated with CX546 and Nifedipine from DIV1 to DIV4. * refers to differences between *Mecp2* null untreated neurons and *Mecp2* null neurons treated with CX546 (*$P$-value < 0.05; **$P$-value < 0.01; ***$P$-value < 0.001); # refers to differences between *Mecp2* null neurons treated with CX546 and null neurons treated with both CX546 and Nifedipine (#$P$-value < 0.05; ##$P$-value < 0.01). Panel J shows the ratio between the expression level of *Kcc2* and *Nkcc1* (expressed as $2^{-\Delta Ct}$). Box plot shows the median (central band), the lower and upper quartiles (boxes), and the max and min values (whiskers) from at least 3 wells obtained from four different embryos. One-way ANOVA was used to compare the effects of the different treatments on null neurons compared to the untreated ones. **$P$-value < 0.01. Sample size: $n \geq 3$ wells from four different embryos.

intracellular level of chloride, which progressively diminishes throughout neuronal development, enabling the transition of GABA responses from excitatory to inhibitory (Ben-Ari, 2002; Ben-Ari *et al*, 2007). The molecular mechanisms driving these events are affected by the lack of Mecp2, resulting in a higher intracellular chloride level in mature null neurons (Tang *et al*, 2016). In line with these findings, the intracellular chloride, imaged at single-cell resolution with MQAE dye (Corradini *et al*, 2018), was higher in null neurons compared to control at each time point we analyzed (the MQAE signal is inversely proportional to the chloride content; #$P < 0.05$, **$P < 0.01$; two-way ANOVA followed by Bonferroni *post hoc* test; Fig 4D and E). Once again, the early exposure of null neurons to CX546 restored an intracellular chloride level comparable to wt neurons (Fig 4F), while no effect was observed in the late time window of treatment (***$P < 0.001$; one-way ANOVA followed by Tukey's *post hoc* test; Fig 4G).

Eventually, we focused on transcription, testing also in primary neurons the ability of CX546 to positively modulate the expression of genes that are typically downregulated in the absence of Mecp2 (Fig 4I). To confirm that CX546 affected transcription through the positive modulation of activity, we included in the analyses samples simultaneously treated with CX546 and the L-type Calcium channels blocker Nifedipine. As expected, the early treatment with CX546 induced the expression of several genes (*Gabbr1, Gap43, Grin1, Kcc2*, and *Tuj1*) that are otherwise downregulated in null neurons (Fig 4H and I). On the contrary, much of these transcriptional effects were not observed in the presence of Nifedipine, thus confirming both the involvement of voltage-gated L-type calcium channels in the mechanism of CX546 action and the ability of increased neuronal activity to ameliorate selected *Mecp2*-dependent transcriptional deficiencies. Importantly, CX546 drove a reduction in the expression of *Rest*, a transcriptional repressor that is generally

upregulated in *Mecp2* null samples and cooperates with Mecp2 in the regulation of *Kcc2* expression (Abuhatzira *et al*, 2007; Tang *et al*, 2016; ***$P < 0.001$, **$P < 0.01$, *$P < 0.05$, #$P < 0.05$; one-way ANOVA followed by Tukey's *post hoc* test). In line with this, we found that CX546, but not CX546 + Nifedipine, increased the ratio of the expression levels of *Kcc2* and *Nkcc1* ($1.8 \pm 0.18$ in CX546-treated samples and $1.1 \pm 0.11$ in CX546 + Nif-treated samples; **$P < 0.01$ one-way ANOVA followed by Tukey's *post hoc* test; panel J). These transcriptional effects fit well with our functional data and suggest a faster rate of maturation of GABA signaling.

### Early treatment with Ampakine ameliorates behavioral deficits of *Mecp2* null mice

Prompted by the rescue effects driven by the early *in vitro* exposure to CX546, we assessed its effectiveness *in vivo*. To this aim, we used the CD1 *Mecp2*^tm1.1Bird^ RTT mouse model focusing on hemizygous ko males (Cobolli Gigli *et al*, 2016). Although RTT mainly affects females, explorative studies are generally performed on mutant male mice that present more robust and consistent phenotypes. *Mecp2* null and wt mice were daily injected subcutaneously with either CX546 (40 mg/kg) or vehicle from P3 to P9 and the validity of the treatment was evaluated starting from P30 testing multiple read-outs (Fig 5A). As expected, the lifespan of CD1 ko mice treated with vehicle was markedly reduced compared to the wt group, showing 50% of survival animals at P80 (Cobolli Gigli *et al*, 2016; Fig 5B). However, when CX546 was injected in the selected early time window, mice showed a significant extension of their lifespan, with 90% of animals being able to survive after P88, an age reached only by a minority of the ko-untreated progeny. Indeed, the average survival of CX546-treated ko animals was P110 (Fig 5B). This effect on lifespan was accompanied by a significant improvement of the

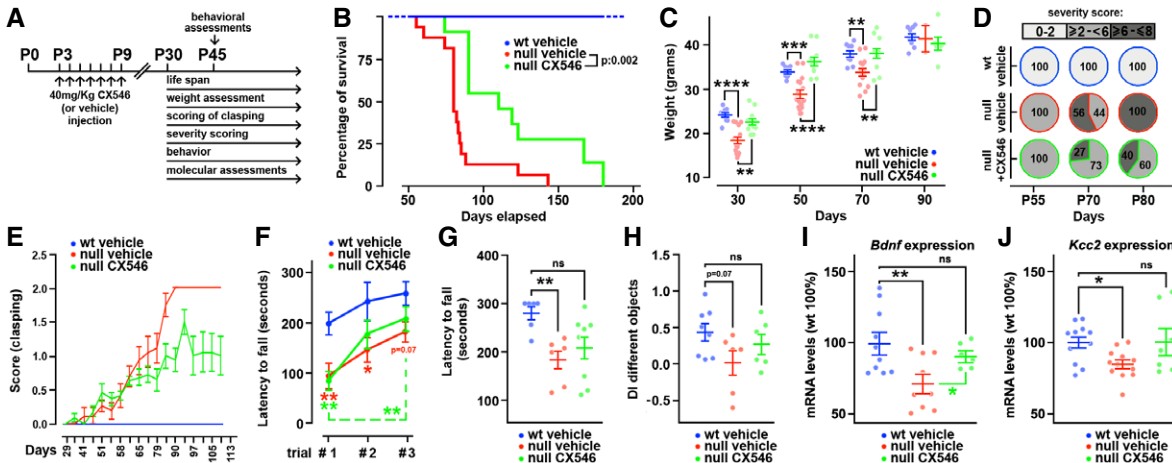

**Figure 5. Early treatment with Ampakine ameliorates behavioral deficits of *Mecp2* null mice.**

A    Schematic representation of the in vivo CX546 administration protocol.

B    Kaplan Mayer test shows an improved lifespan of *Mecp2* null CX546-treated animals compared to *Mecp2* null control animals. Sample size: wt = 11, ko = 16, ko treated with CX546 = 11. Gehan–Breslow–Wilcoxon test was used to compare the two groups. **P-value < 0.01.

C    The body weight of mice was assessed at four different time points (P30, P50, P70, and P90). Two-way ANOVA indicated a significant effect for both time ($F_{(1, 813, 47, 13)}$ = 421.1 and P < 0.0001) and treatment ($F_{(2, 33)}$ = 7.540 and P < 0.0001). Tukey's multiple comparison test was used to compare within each time point the effect of the CX546 treatment on *Mecp2* null mice. **P-value < 0.01; ****P-value < 0.0001. Values are represented as average ± SEM. Sample size: n = 10 wt and n = 27 ko (16 treated with vehicle and 11 treated with CX546). Each dot represents a single animal.

D, E    Severity scores were calculated at different time points (P55, P70, P80 in D). The score for hind limbs clasping was assessed twice a week starting from P30 (E). Sample size for D and E: n = 10 wt treated with vehicle and n = 20 ko (9 treated with vehicle and 11 treated with CX546). Data are represented as mean ± SEM.

F, G    Line graph (F) shows mice motor learning assessed on the accelerating rotarod during three consecutive trials. Two-way ANOVA indicated a significant effect of time ($F_{(2, 21)}$ = 5.490 and P = 0.0121) and genotype ($F_{(2, 33)}$ = 16.33 and P < 0.0001). Bonferroni multiple comparison test highlighted a significant increase in the latency to fall between trial 1 and trial 3 only for null mice treated with CX546. *P-value < 0.05; **P-value < 0.01. The scatter plot (G) shows latency to fall (in seconds) on the accelerating rotarod in the last trial. One-way ANOVA was used to compare ko mice and ko-treated mice to the wt group. The Tukey's multiple comparison test highlighted a significant reduction in the latency to fall between wt and ko but not between wt and ko treated with Ampakine; **P-value < 0.01. Values are represented as average ± SEM. Sample size: n = 6 wt treated with vehicle and n = 14 ko (6 treated with vehicle and 8 treated with CX546). Each dot represents a single animal.

H    The graph shows the assessment of the discrimination index (DI) between two different objects for each mouse during the last trial of a Novel Object Recognition test. One-way ANOVA was used to compare ko mice and ko-treated mice to the wt group. Tukey's multiple comparison test highlighted a P-value of 0.07 between wt and ko treated with vehicle but not between wt and ko treated with CX546. Data are represented as mean ± SEM. Sample size: n = 8 wt treated with vehicle n = 12 ko (6 treated with vehicle and 6 treated with CX546). Each dot represents a single animal.

I, J    The graphs show the expression level of *Bdnf* (I) and *Kcc2* (J) in cortical samples obtained from P45 animals. One-way ANOVA was used to assess difference in the expression of the two selected genes between wt, ko e CX546-treated ko animals. For *Bdnf* expression (I), Tukey's multiple comparison test highlighted significant differences between both wt and ko and ko and ko treated with CX546 but no difference between wt and ko treated with CX546 (*P-value < 0.05; **P-value < 0.01). For *Kcc2* expression (J), Tukey's multiple comparison test highlighted significant differences between wt and ko but no difference between wt and ko treated (*P-value < 0.05). Data are represented as values ± SEM. Sample size for I: n = 10 wt, n = 9 ko, and n = 6 ko treated with CX546. Sample size for J: n = 11 wt, n = 12 ko, and n = 7 ko treated with CX546. Each dot represents a single animal.

overall health of KO-treated mice. First, the thoroughly described reduction in body weight of *Mecp2* null mice was markedly recovered after the treatment with CX546 at all the analyzed time points (****P < 0.001, **P < 0.01, *P < 0.05; two-way ANOVA followed by Bonferroni *post hoc* test; Fig 5C). By means of the scoring system typically used to assess the worsening of the health conditions of null mice (Guy *et al*, 2001), we noticed a marked difference in the severity between untreated and treated ko animals during the overt phase of the pathology (Fig 5D). At P70, in fact, while 56% of untreated ko mice exhibited a severe RTT score (>6), only 27% of CX546-treated mice felt in this highly symptomatic range. Such difference was even more evident at P80, when we detected that all the untreated ko mice felt in the highest severity score, as opposed to only 40% of the CX546-treated mice. Among the phenotypes observed during the scoring assessment, hind limb clasping, one of the classical parameters used to assess neurological defects in RTT animal models, was ameliorated after CX546 treatment (Fig 5E).

Next, we assessed the effectiveness of the treatment in ameliorating defects typically displayed by RTT mouse models in locomotor activity, coordination, and spatial memory. The CX546 treatment significantly ameliorated the performances of P45 mice on the accelerating rotarod test, as shown by the assessment of learning ability (through three successive trials; Fig 5F) and motor coordination (last trial; **P < 0.01, *P < 0.05; two-way ANOVA followed by Bonferroni *post hoc* test; Fig 5G). At the same age, we tested the animals in the Novel Object Recognition test (NOR) proving that the early exposure to Ampakine improved the ability of ko animals to discriminate between two different objects (**P < 0.01, *P < 0.05; two-way ANOVA followed by Bonferroni *post hoc* test; Fig 5H and Appendix Fig S3). Notably, CX546 treatment did not affect both physical and behavioral performance of wt-treated animals (Appendix Fig S3).

Interestingly, and in good accordance with Ampakine effects (Lauterborn *et al*, 2003; Lynch & Gall, 2006), the amelioration of

*Mecp2* null animal conditions was accompanied by rescued transcriptional levels of *Bdnf* in P45 null cortices ($P < 0.01$, *$P < 0.05$; one-way ANOVA followed by Tukey's *post hoc* test; Fig 5, panel I). Further, and in line with our *in vitro* results, CX546 treatment rescued the *Kcc2* transcriptional defect (Fig 5, panel J). To be noticed, CX546 did not affect *Bdnf* and *Kcc2* transcription in wt-treated cortices (Appendix Fig S3).

## Discussion

Recent preclinical and clinical studies demonstrate that besides the importance of Mecp2 in maintaining neuronal structures (Guy *et al*, 2007; McGraw *et al*, 2011; Cheval *et al*, 2012; Nguyen *et al*, 2012), its deficiency strongly affects embryonic and early postnatal development (Ip *et al*, 2018), thus long before the full outbreak of symptoms (Cosentino *et al*, 2019; Zhang *et al*, 2019). We contributed to this novel perspective by demonstrating that *Mecp2* null neurons display aberrant transcriptional profiles, reduced responsiveness to *stimuli* and poor morphology already during early corticogenesis (Bedogni *et al*, 2016; Cobolli Gigli *et al*, 2018). Based on the hypothesis of a crucial role played by neuronal activity in the establishment of mature neuronal networks (Spitzer, 2006), we investigated whether the reduced responsiveness of *Mecp2* null neurons participates to their poor transcriptional and morphological features that would further contribute to the overall immaturity of the networks. We thus tested the rescue potential of an early enhancement of activity in *in vitro* and *in vivo Mecp2* null models.

To enhance activity, we used the Ampakine CX546, a positive modulator of AMPA glutamatergic receptors that upon glutamate binding delays deactivation and desensitization (Lynch & Gall, 2006). The choice of CX546 was based on its ability to stimulate dendritic outgrowth and maturation of *in vitro* differentiating neurons (Schitine *et al*, 2012) and was further supported by the fact that CX546 was already successfully used in preclinical studies on RTT, although in later time windows compared to this study (Ogier *et al*, 2007) and focusing on a different pathological mechanism (Degano *et al*, 2014). Our data demonstrate that the maturation of *Mecp2* null neurons is sensitive to depolarization. In fact, by exposing *Mecp2* null neurons to KCl we rescued morphological parameters, which was in line with the beneficial effects driven by CX546 on morphology, transcriptional levels, and network functions. By enhancing activity, we thus restored each component of the already described feed-forward cycle that ensures neuronal maturation (Spitzer, 2006). In good accordance with our hypothesis, the early treatment resulted more effective compared to the late one possibly because during early time windows neuronal networks are more plastic compared to later stages, when changes are restrained by advanced differentiation. In fact, CX546 treatment increased the levels of expression of *Gap43* and *Tuj1*, two structural proteins that are involved in synaptic plasticity and growth cones stability (Benowitz & Routtenberg, 1997; Korshunova & Mosevitsky, 2010; Tischfield & Engle, 2010). On the same line, CX546 exposure increased transcription of the neurotrophin *Bdnf*, a master regulator of plasticity (Lauterborn *et al*, 2003; Simmons *et al*, 2009) thoroughly associated with MeCP2 deficiency (Li *et al*, 2017; Renthal *et al*, 2018). From a functional perspective, CX546 rescued the responsiveness of null neuronal networks to *stimuli*, while

decreasing their intracellular $Cl^-$ levels, which suggests a restored responsiveness to GABA. This result fits with the recovered transcriptional levels of *Kcc2*, the neuron-specific $K^+$-$Cl^-$ co-transporter (KCC2) that is crucial for GABA signaling maturation (Ben-Ari, 2002; Ben-Ari *et al*, 2007) and its normalization in Rett syndrome currently represents a promising therapeutic approach (Tang *et al*, 2016; Hinz *et al*, 2019; Tang *et al*, 2019; Lozovaya *et al*, 2019). Importantly, Nifedipine, a voltage-gated calcium channels blocker, restricted the observed transcriptional rescues, therefore proving the importance of calcium signaling in controlling many transcriptional programs involved in neuronal development (Rosenberg & Spitzer, 2011). Our data thus support the concept of a deficit of calcium-dependent maturation mechanisms in neurons lacking *Mecp2* (Bedogni *et al*, 2016).

A further validation to our hypothesis derived from the benefits we detected in *Mecp2* null animals treated from P3 to P9 with CX546. Indeed, we demonstrated that such a short and early treatment delays the progression of the disorder. In fact, ko-treated mice manifested a significantly prolonged life span, an amelioration of motor and cognitive functions (evaluated 30 days after the treatment) and a delayed progression of typical phenotypes that were still evident 70 days after the last administration with CX546.

The benefits produced by early CX546 treatment were further confirmed by the rescued transcriptional levels of *Bdnf* and *Kcc2* observed in the cortex of P45 null animals, which reinforces our *in vitro* results and further suggests the role of GABA signaling in mediating ampakine-positive effect. Overall our *in vivo* results suggest a remarkably long-lasting effect of the treatment, thus reinforcing the role of early changes in plasticity on the functions of *Mecp2* null neuronal networks. While the alteration of the already described feed-forward cycle during early development does not produce overt symptoms, such alterations likely concur to the establishment of an improper program of gene expression that takes part to the outbreak of overt symptoms and the typical *Mecp2* null network dysfunctions later in life (Dani *et al*, 2005; Durand *et al*, 2012; Nelson & Valakh, 2015; Sceniak *et al*, 2016). We thus propose the existence of an early molecular phase of Rett syndrome during which the initial developmental trajectory undertaken by maturing null networks deranges from physiology. We believe that a thorough comprehension of the mechanisms playing a pathogenic role during this phase will likely reveal novel therapeutic strategies.

The importance of targeting early neuronal maturation has already been implied by previous preclinical and clinical studies (Marín, 2016). Indeed, the injection of a truncated form of IGF1 in pre-symptomatic (P15) *Mecp2* null and heterozygous mice rescued neuronal morphology, synaptic density, and network plasticity (Tropea *et al*, 2009). Further, in clinical trials, IGF1 (Trofinetide) revealed more effective when administered at pediatric ages compared to adolescence/adult (Banerjee *et al*, 2019). Similarly, the efficacy of the $GABA_B$ receptor agonist Arbaclofen in the cure of Fragile-X syndrome symptoms resulted strictly dependent to the timing of treatment (Berry-Kravis *et al*, 2018). Accordingly, in a mouse model of epileptic encephalopathy, GABAergic tone modulators ensured phenotypic rescue only when administered within the first 2 weeks of life (Marguet *et al*, 2015).

RTT is generally considered a reversible condition and the ability of CX546 to recover *Bdnf* expression and LTP in the adult *Mecp2* null brain has already been suggested (Ogier *et al*, 2007). However,

these studies did not assess the persistence of the effects and the behavioral domains that benefitted from the treatment. Although we suggest that an early enhancement of activity might have a higher positive effect than later ones, such paradigm of CX546 administration may be impractical for translational approaches. Thus, in the future we will apply longitudinal studies and test different time windows of treatment to gain preclinical insights. Further, we will assess whether the *in vivo* timing of CX546 exposure should be enlarged to ensure wider and possibly stronger rescue effects. Eventually, we will test the benefits of repeating the treatment in order to ensure the preservation, rather than the establishment alone, of functional neuronal networks; promising data will be confirmed in *Mecp2* heterozygous females, as requested for any study testing a novel therapeutic approach for RTT (Katz *et al,* 2016).

All in all, our study suggests that overt symptoms are the final outcome of a pathogenic process that likely starts very early during development, implying the importance of re-evaluating the definition of the "pre-symptomatic" phase of Rett syndrome (Bedogni *et al,* 2016; Cosentino *et al,* 2019; Zhang *et al,* 2019).

# Materials and Methods

### Animals and tissues

The *Mecp2* mouse strain was originally purchased from Jackson Laboratories (B6.129P2(C)-*Mecp2*^tm1.1Bird^/J) and transferred on a CD1 genetic background. The phenotypes affecting these animals have been previously described (Guy *et al,* 2001; Cobolli Gigli *et al,* 2016). Mouse genotypes were determined through PCR on genomic DNA purified from tail biopsy obtained within the second and the third week of life (Cobolli Gigli *et al,* 2016). Time pregnant females were generated by crossing overnight wt CD1 males with *Mecp2*^−/+^ heterozygous females; the day of vaginal plug was considered E0.5. For RNA processing, brains were collected from anesthetized mice (Avertin, Sigma-Aldrich), dissected cortices were then rapidly frozen on dry ice and stored at −80°C. All procedures were performed in accordance with the European Community Council Directive 86/609/EEC for care and use of experimental animals; all the protocols were approved by the Italian Ministry for Scientific Research and by the local Animal Care Committee.

### Neuroprecursors and neurons cultures

Cerebral cortices were dissected from E15.5 mouse embryos and pulled after genotyping. Generation and maintenance of neurosphere cultures were performed as already described (Cobolli Gigli *et al,* 2018; Fig 1A). To induce differentiation, neurospheres were dissociated in single-cell suspensions and plated on Matrigel-coated surfaces. Until adhesion, cells were grown in complete medium (DMEM/F12, 0.6% glucose, 1% L-glutamine, 1% Pen/Strep, 4 μg/ml heparin, 1× hormone mix, 20 ng/ml epidermal growth factor (EGF), 10 ng/ml basic fibroblast growth factor (bFGF)). EGF and bFGF were withdrawn, respectively, 2 and 3 days after plating before the addition of fetal bovine serum (FBS 2%). This last passage was considered DIV0. Generally, cells were plated at a density of 26,000 cells/cm². To enhance activity, CX546 (Tocris Bioscience) was dissolved in DMSO and diluted 1:10 in water to obtain the working solution. The drug was added daily for 4 days directly to the medium from either DIV3 or DIV7 reaching a final concentration of 20 μM. An equal volume of 10% DMSO in water was added to controls. CX546 was washed out following the last administration by replacing the cell medium.

Primary neuronal cultures were prepared from E14.5 mouse cerebral cortices (Gandaglia *et al,* 2018). 15,000 neurons/well were plated on poly-L-lysine (0,1 mg/ml; Sigma-Aldrich) coated 96 multiwells (Greiner) and cultured in Neurobasal medium supplemented with B27, L-Glutamine, and Pen-Strep. CX546 (10 μM) or corresponding volumes of DMSO were directly added to the medium at DIV1 and DIV3 for the early treatment and at DIV7 and DIV9 for the late treatment. Nifedipine (10 μM; Tocris Bioscience) was added at the same time of CX546 only within the early time window at DIV1 and DIV3. Drugs were then washed out by changing the medium at DIV5 or DIV11.

### Cell immunofluorescence

Cells were fixed with 4% PFA for 20 min, rinsed with PBS, and incubated for 1 h in blocking solution (10% horse serum, 0.1% Triton X-100). Cells were then incubated overnight at 4°C with primary antibodies diluted in blocking solution at the concentrations reported in Appendix Table S2. After PBS washing, cells were incubated for 1 h at room temperature in secondary antibodies (1:500 in blocking solution; Molecular Probes). Before mounting, cells were rinsed with PBS and nuclei were counterstained with DAPI (Invitrogen). An upright Nikon microscope was used for acquisitions, while ArrayScan XTI HCA reader (Thermo Fisher Scientific) was used for all the high-throughput acquisitions. For the measurement of Neun intensity, images were all acquired with the same setting and then the signal was manually measured with ImageJ. Excel and Prism were used to elaborate data.

### RNA purification, cDNA synthesis, and quantitative PCR

#### NPC and primary neurons for qPCR Dynamic Array microfluidic chip assays

Cells were rinsed once with PBS and lysed in Purezol™ (300 μl for roughly 30,000 cells; Bio-Rad). RNA was isopropanol precipitated and treated with DNAse I (Sigma-Aldrich), re-extracted with phenol-chloroform and isopropanol precipitated over night at −20°C. RNA pellets were dissolved in 10 μl of TE buffer. RNA concentration and quality were checked using Bioanalyzer RNA 6000 Nano chips (Agilent). 50 ng of total RNA was reverse-transcribed using Superscript IV Reverse Transcriptase (Thermo Fisher). 1 ng of cDNA for each sample was pre-amplified using a 0.2X pool of 74 primers (TaqMan; Thermo Fisher; Appendix Table S1) for 18 cycles using PreAmp Master Mix (Fluidigm) to enable multiplex sequence-specific amplification of targets. Pre-amplified cDNAs were then diluted and assessed using a 96 × 96 qPCR Dynamic Array microfluidic chip (Fluidigm) following the manufacturer's instructions. Baseline correction was set on Linear (Derivative) and Ct threshold method was set on Auto (global). To normalize, for each sample a "pseudogene" (D'haene *et al,* 2012) was obtained by averaging the Ct value of each gene depicted in Appendix Table S1 except for *Mecp2*. Each sample was then normalized against its "pseudogene." Excel, Prism, and R were used to elaborate data.

### DIV22 NPCs for quantitative RT–PCR

Cells were rinsed once with PBS and lysed in Purezol™ (300 μl for roughly 30,000 cells; Bio-Rad). RNA was isopropanol precipitated and treated with DNAse I (Sigma-Aldrich), re-extracted with phenol-chloroform and isopropanol precipitated over night at −20°C. RNA pellets were dissolved in 10 μl of TE buffer. RNA quantity was checked using NanoDrop (Thermo Fisher Scientific). 100 ng of RNA was then retro-transcribed using the RT$^2$ First Strand kit (Qiagen). qRT–PCR results depicted in the Fig EV2 were performed using SYBR green (Life Technologies) as fluorescent dye. Primers used for the analyses are reported in Appendix Table S1. Excel (Microsoft) and Prism (GraphPad) were used to elaborate data.

### Tissues for quantitative RT–PCR

Cortices were lysed in 500 μl of Purezol™ (Bio-Rad). RNA was isopropanol precipitated and treated with DNAse I (Sigma-Aldrich), re-extracted with phenol-chloroform and isopropanol precipitated over night at −20°C. RNA pellets were dissolved in 20 μl of Nuclease free water (Sigma-Aldrich). RNA quantity was measured using NanoDrop. 1,000 ng of RNA was then retro-transcribed using the RT$^2$ First Strand kit (Qiagen). qRT–PCR results depicted in Fig 5 and in the Appendix Fig S3 were performed using SYBR selected master mix (Applied Biosystems) as fluorescent dye. Primers used for the analyses are reported in Appendix Table S1. Excel (Microsoft) and Prism (GraphPad) were used to elaborate data.

### Western blots

Neurons from 6-multiwell plates were lysed in ice-cold lysis buffer (50 mM Tris–HCl, pH 7.4, 150 mM NaCl, 5 mM EDTA, 1% NP-40, 1× complete EDTA-free protease inhibitor cocktail (Roche) and 1× PhosSTOP (Roche)) for 30 min on ice and then centrifuged for 30 min at 10,000 $g$ (4°C). Protein concentrations were measured by the bicinchoninic acid (BCA) assay kit (Thermo Fisher Scientific). Equal amounts of protein lysates were separated on TGX Stain-Free gel (Criterion 12 wells 4–15%; Bio-Rad). Before transfer, a stain-free gel image was acquired by ChemiDoc Touch Imaging System (Bio-Rad) and used to quantify results. Proteins were blotted on a nitro-cellulose membrane (Trans-blot Turbo Nitrocellulose Transfer Packs; Bio-Rad) using the Trans-blot SD semidry apparatus (Bio-Rad). Membranes were incubated 1 h in blocking solution (Tris-buffered saline containing 5% nonfat milk and 0.1% Tween-20, pH 7.4) and then incubated overnight (4°C) with the primary antibody diluted in blocking solution. The following primary antibodies were used: anti-AKT (4685; Cell Signaling; 1:2,000) and anti-phospho-AKT (Ser473) (4060; Cell Signaling; 1:2,000). After 3 washes in TBS-T, blots were incubated with the appropriate HRP-conjugated secondary antibody (Peroxidase-conjugated AffiniPure Goat anti-rabbit IgG (H + L), Jackson ImmunoResearch) for 1h at room temperature. Immunocomplexes were visualized using the ECL substrates kit from Cyanagen and the Bio-Rad ChemiDoc™ System. Quantification of bands was performed using the Image Lab 5.2.1 Software (Bio-Rad).

### MEA recordings

Neuroprogenitors were plated at a density of 85,000 cells/well onto 12-well planar MEA chips (Axion BioSystems, Atlanta, GA), comprising 64 electrodes/well. Spontaneous extracellular activity was recorded with Axion Biosystems software (Axion Integrated Studio), by individually setting a voltage threshold for each channel equal to seven times the standard deviation of the average root mean square (RMS) noise level. Single extracellular APs were detected by threshold crossing of 200 Hz high-pass filtered traces. Spike detection was carried out using the Axion BioSystems software Neural Metric Tool. Evoked extracellular activity was measured after addition of 100 μM 4-AP directly to cell medium. An electrode was defined active when it registered at least 2 spikes/min. Weighted mean firing rate (WMFR) was expressed as mean number of spikes per second calculated on the active channels of the network. Bursts within single channels were identified by applying an interspike interval (ISI) threshold algorithm (Chiappalone et al, 2005) by defining bursts as collections of a minimum number of spikes ($N_{min}$ = 5) separated by a maximum interspike interval ($ISI_{max}$ = 100 ms). Only wells showing at least three active electrodes were included in the analysis.

### Automated analysis of calcium transients

Primary cortical neurons were loaded with 2 μM Fluo-4 (Invitrogen) in KRH (Krebs'–Ringer's–HEPES containing (in mM): 125 NaCl; 5 KCl; 1.2 MgSO$_4$; 1.2 KH$_2$PO$_4$; 25 HEPES; 6 glucose; 2 CaCl$_2$; pH 7.4) for 30 min at 37°C and then washed once with the same solution. Stimulation was performed automatically by using the liquid handling system of the ArrayScan XTI HCA Reader (Thermo Fisher Scientific). To stimulate, one dose of NMDA (100 μM at the rate of 50 μl/s) was added while images were digitally acquired with a high-resolution camera (Photometrics) through a 20× objective (Zeiss; Plan-NEOFLUAR 0.4 NA). Hoechst fluorescence was imaged as well. 70 frames were acquired at 1 Hz with 40 ms exposure time for Fluo-4 and 25 ms exposure time for Hoechst. At least 8 baseline images were acquired before stimulation. The analysis was done with HCS Studio software using SpotDetector bioapplication (Thermo Fisher Scientific). Hoechst-positive nuclei were identified and counted, and the mean intensity of the Fluo-4 signal was measured in the cell body area of each cell; background intensity was measured and subtracted from the mean intensity. Only cells with neuronal morphology were included in the analysis. Calcium responses were measured as $\Delta F/F_0$.

### Chloride imaging recordings

Intracellular chloride measurements were performed using the N-(ethoxycarbonylmethyl)-6-methoxyquinolinium bromide (MQAE) (Biotium) chloride sensor. Briefly, cortical neurons at 14 DIV from wt and Mecp2 null embryos were loaded with 5 μM MQAE for 45 min at 37°C in culture medium. Coverslips were then rinsed with external solution (Krebs'–Ringer's–HEPES (KRH): 125 mM NaCl, 5 mM KCl, 1.2 mM MgSO$_4$, 1.2 mM KH$_2$PO$_4$, 2 mM CaCl$_2$, 6 mM glucose, and 25 mM HEPES–NaOH), pH 7.4 and transferred to the recording chamber for acquisitions.

Olympus IX81 inverted microscope, 20× dry objective (Olympus, UPLFLN NA 0.5) provided with MT20 widefield source and control system with excitation 340 nm and emission filter centered at 500 nm were used. EXcellence RT software (Olympus) was used to set up experiments and samples' recordings. For each sample, a

minimum of eight fields of interest (mean of cells observed for field, 20) were chosen to perform live imaging recordings. Offline analyses were then made after defining ROIs centered to cell bodies and measuring mean MQAE intensity during the time recording window.

## Morphological analyses

Morphological analyses were performed on images acquired with a Nikon fluorescence microscope using a 20× objective. A binary mask was created for each Tuj1-positive cell using Adobe Photoshop by an observer blind to the genotype and the treatment. Sholl analyses and soma size were performed using the dedicated plugin of ImageJ. The smallest circle was positioned at 5 μm from the soma, while the largest at 300 μm. The distance between each circle was 10 μm. The total dendritic length (TDL) and maximal dendritic length (MDL) were evaluated using NeuronJ, a plugin of ImageJ. For each neuron, the length of each dendrite (Lm) was measured and the sum of such values defined the total dendritic length (TDL). The polarization value was calculated as the ratio between the maximal dendritic length and the average dendritic length. A ratio equal or above 2 defines a polarized neuron; a culture is defined polarized when at least 60% of neurons within it are polarized (Horton et al, 2006).

## Cell viability assays

Cell viability was assessed through MTT assay (3-(4,5-dimethylthiazol-2-yl)-2,5-diphenyltetrazolium bromide; Sigma-Aldrich). MTT (4 mg/ml) was diluted 1:10 in DMEM-F12 and added to each well after medium withdrawal; plates were then incubated at 37°C for 4 h. Next, the solution was replaced by 100 μl of DMSO, and the colorimetric signal assessed with a spectrophotometer (570 nm).

## In vivo drug administration

Ampakine CX546 40 mg/kg was administered daily via a subcutaneous injection for 7 days (starting from P3) at the same time. The drug was solubilized in ethanol (100%) and diluted 1:100 in saline to obtain the working concentration. For the vehicle treatment, ethanol (100%) was diluted 1:100 in saline.

## Neurobehavioral characterization, phenotypic scoring, and molecular assessments

Weight and phenotypic scoring assessments were evaluated twice a week starting from P30. Severity score, typically used in RTT phenotypic assessments (Guy et al, 2001; Cobolli Gigli et al, 2016), was used to group animals into three severity classes: absence of phenotypes (0–2; light gray), mild phenotypes (≥ 2 to < 6; medium gray), and severe phenotypes (≥ 6 to ≤ 8; dark gray), as described in Patrizi et al (2016). The percentage refers to the number of mice falling in one of the three severity classes over the total number of animals per group. A total of 25 wt and 25 ko mice were used for lifespan and phenotypic scoring. To be noticed, mice that rapidly lost weight were euthanized for ethical reasons. The day of the sacrifice was considered as the endpoint of lifespan assessment.

For molecular assessments (qRT–PCR on cortical samples), mice were sacrificed at P45; a total of 16 wt and 16 ko mice were used for this experiment. To avoid any bias, an investigator blind to the genotypes and treatments of tested animals performed all the analyses.

## Behavioral assessment

Animals were maintained on an inverted 12 h light/darkness cycle at 22–24°C. A total of 18 wt animals and 14 ko animals were involved in the behavioral assessment. To avoid any bias, an investigator blind to the genotypes and treatments of tested animals performed all the analyses.

### Rotarod

Mice were assessed on an accelerating rotarod (Ugo-Basile, Stoelting Co.). The test was carried out in 3 days, 2 days of training and 1 day of test. Each session consisted of three trials and each trial lasted 5 min. Revolutions per minute (rpm) were set at an initial value of 4 with a progressive increase to a maximum of 40 rpm. Each trial ended when the mouse fell down or after 5 min. Latency to fall was measured by the rotarod timer. Data were plotted and subjected to statistical analysis with GraphPad Prism.

### Novel Object Recognition (NOR)

The test was performed in an open square arena of 50 × 50 cm and consisted of three different sessions. On day 1, during the first session, mice were allowed to habituate with the arena for 10 min. On day 2, mice underwent the training phase (5 min) in which they were allowed to explore two identical objects. Object investigation was defined as time spent sniffing the object when the nose was oriented toward the object and the nose-object distance was 2 cm or less. Time spent sniffing two identical objects during the familiarization phase confirmed the lack of an innate side bias. The total time spent by the mice to explore the two objects in the cage was also measured; if this was < 30 s (10% of the total time), the mouse was not included into the final test (Appendix Fig S5). Finally, on day 3, mice were tested (5 min) using two different objects. Recognition memory, defined as spending significantly more time sniffing the novel object than the familiar object, was calculated as Discrimination Index (DI): difference between the exploration time for the novel object and the familiar object, divided by total exploration time. The sessions were recorded with the video tracking software EthoVision XT (Noldus).

## Statistical analysis

Statistical analysis and plotting of data were performed with GraphPad Prism. Student's t-test was used for the statistical analysis when wt were compared to Mecp2 null mutant samples. One-way ANOVA was used to statistically compare the effects of early and late addition of CX546, the effect of CX546 or CX546 plus Nifedipine on Mecp2 null neurons and the effects of ampakine treatment on mice. Data from the experiments on wt and null NPCs or neuronal cultures at different DIVs, from the assessment of mice weight and from the assessment of the exploration rate were analyzed by two-way ANOVA. When there was a significant effect of treatment or genotype, or a significant interaction between the variables (one-

**The paper explained**

**Problem**

*MECP2* mutations generate a number of neurological conditions among which the most frequent is Rett syndrome (RTT). While RTT signs become overt after an apparently asymptomatic phase, early developmental defects have been detected, but their contribution to the pathogenesis of RTT is still not understood. The description of such early events could elucidate the pathogenesis of RTT and help the proposition of new therapies, as only supportive strategies are currently available for RTT patients.

**Results**

We show that the lack of Mecp2 reduces the ability of neurons to respond to stimuli and alters their transcription and morphology. In normal conditions, neuronal activity drives transcriptional and morphological changes that support, in a feed-forward cycle, maturation. We thus enhanced activity through the administration of the Ampakine CX546 in cellular and animal models of RTT. Our data show the effectiveness of this strategy, which proves more efficient if applied during early stages of maturation, when brain circuits are more prone to plasticity changes.

**Impact**

By showing the rescue effects produced by enhancing activity during early stages of maturation, we highlight the contribution of defective mechanisms of development to the phenotypes typically displayed by RTT models. Our data show that Ampakine administration during early development produces beneficial effects that persist long after the end of the treatment.

way or two-way ANOVA), appropriate *post hoc* test was applied. A *P*-value of 0.05 was considered significant. A list of exact *P*-values is provided as Appendix Table S3. Possible outliers within an experimental group were identified with Grubb's test and discarded from the final analysis.

Each culture wells from *in vitro* analysis and mice from *in vivo* experiments were randomly assigned to treatments, and for all the experiments reported in the manuscript, the investigators were blinded to the treatments and genotypes during data acquisition and analyses.

## Data availability

This study includes no data deposited in external repositories.

**Expanded View** for this article is available online.

## Acknowledgements

We are grateful to ProRett Research, an Italian association that daily provides us with both strive and funding to carry on our study on Mecp2 and Rett syndrome. Lejeune Foundation (Paris, France) and Fondazione Umberto Veronesi (Milan, Italy) provided additional funding to FBedo. AIRC (Associazione Italiana per la Ricerca sul Cancro; Grant no. IG2016-ID18575) and ERC (European Research Council, Consolidator Grant no. 617978) provided additional funding to MP. Compagnia di San Paolo Torino (9344), Ministero della Salute Ricerca Finalizzata (GR-2016-02363972), and EU Era-Net Neuron 2017 "Snareopathies" provided additional funding to FBen. The team of the microscopy facility at the San Raffaele Hospital (ALEMBIC) produced all the imaging data described in this study. We thank the facility of Mouse Behavior studies at the San Raffaele Hospital for supplying the equipment required for the behavior tests. We thank Dr. Marzia Indrigo for her contributions in both behavioral and molecular assessments.

## Author contributions

LS, FB, and NL conceived the study and produced the manuscript. LS, GDR, GD, CCG, MC, FM, DP, MDS, and PC produced experimental data. MP, FB, and FC contributed to the interpretation of parts of the results.

## Conflict of interest

The authors declare that they have no conflict of interest.

## For more information

- https://prorett.org
- https://www.fondationlejeune.org

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
