## [Review Process File · EMBO Molecular Medicine]

The enhancement of activity rescues the establishment of *Mecp2* null neuronal phenotypes

Linda Scaramuzza, Giuseppina De Rocco, Genni Desiato, Clementina Cobolli Gigli, Martina Chiacchiaretta, Filippo Mirabella, Davide Pozzi, Marco De Simone, Paola Conforti, Massimiliano Pagani, Fabio Benfenati, Fabrizia Cesca, Francesco Bedogni, Nicoletta Landsberger

DOI: 10.15252/emmm.202012433

Corresponding authors: Nicoletta Landsberger (nicoletta.landsberger@unimi.it) , Francesco Bedogni (bedognif@cardiff.ac.uk)

Review Timeline:

Submission Date:	4th Apr 20
Editorial Decision:	4th May 20
Revision Received:	22nd Nov 20
Editorial Decision:	24th Dec 20
Revision Received:	24th Jan 21
Accepted:	27th Jan 21

Editor: Zeljko Durdevic

Transaction Report:

4th May 2020

Dear Prof. Landsberger,

Thank you for the submission of your manuscript to EMBO Molecular Medicine. We have now heard back from the three referees who agreed to evaluate your manuscript. As you will see from the reports below, the referees acknowledge the interest of the study. However, they raise some concerns that should be addressed in a major revision of the present manuscript. Addressing the reviewers' concerns in full will be necessary for further considering the manuscript in our journal.

Acceptance of the manuscript will entail a second round of review. Please note that EMBO Molecular Medicine encourages a single round of revision only and therefore, acceptance or rejection of the manuscript will depend on the completeness of your responses included in the next, final version of the manuscript. For this reason, and to save you from any frustrations in the end, I would strongly advise against returning an incomplete revision.

We realize that the current situation is exceptional on the account of the COVID-19/SARS-CoV-2 pandemic. Therefore, please let us know if you need more than three months to revise the manuscript.

I look forward to receiving your revised manuscript.

Yours sincerely,

Zeljko Durdevic

***** Reviewer's comments *****

Referee #1 (Remarks for Author):

This study by Scaramuzza et al sets out to determine whether enhancing neuronal activity during early brain development prevents the onset of Rett syndrome-associated molecular and cellular phenotypes. Working in an in vitro system, the authors demonstrate that Mecp2-null neurons derived from neural progenitor cells from mouse embryonic cortex exhibit deficits in neuronal activity, transcription of a select set of genes, and dendritic morphology. To test whether these effects could be attenuated by pharmacologically increasing excitatory neuronal activity, they

treated progenitor-derived neurons with ampakine CX546. Treating the neurons at two different times resulted in variable effects on transcription and neuronal morphology, with greater effects achieved in response to earlier treatment. The authors then extended these findings, first to a *Mecp2*-null primary neuronal culture system, and then in vivo with *Mecp2*-null mice, demonstrating that ampakine treatment resulted in increased survival, decreases in phenotypic severity score, and rescue of behavioral deficits in two different assays.

The premise of this study is quite novel. Rett syndrome (RTT) is caused by loss of MeCP2, a nuclear protein that binds methylated DNA. Loss of MeCP2 leads to widespread transcriptional changes due to a yet-to-be-defined role of MeCP2 in the regulation of gene expression. These gene expression alterations are thought to be the primary drivers of RTT pathogenesis, leading to the cellular, circuit, and behavioral abnormalities associated with RTT in adulthood. This study, on the other hand, argues that there exists an "early molecular phase" of RTT and that impaired neuronal activity in early development may drive or promote the pathogenesis of RTT, resulting in transcriptional changes that lead to other RTT-like phenotypes. In addition to its interesting hypothesis, this study is well-conducted in general. The authors employed a previously established in vitro system and validated molecular and cellular deficits shown in *Mecp2*-null neurons. Additionally, the authors follow two drug treatment schedules with multiple phenotypic readouts. The experiments are conducted and analyzed in a statistically rigorous manner, with appropriate numbers of biological replicates and the correct statistical tests. Several questions/weaknesses, as outlined below, if appropriately addressed, would significantly improve the study and warrant publication in EMBO Molecular Medicine.

1. The authors use the term "transcriptional maturity" to describe the molecular signature of *Mecp2*-null neurons and argue that enhancing excitatory neuronal activity improves transcriptional maturity and attenuates cellular phenotypes. Based on the gene expression studies presented in Figure 1, the authors conclude that there is a deficit in transcriptional maturation due to changes in the expression of markers of progenitor cells at earlier time points and changes in the expression of markers of post-mitotic neurons at later time points. This does not necessarily imply a deficit in transcriptional "maturation" - to claim there is a deficit in maturation it would be necessary to show that *Mecp2*-null cells are "immature" at later stages, with expression profiles resembling an earlier stage (which does not seem to be the case based on the PCA plots). However, this could reflect deficits in transcriptional "identity" at different timepoints, particularly given the recent findings that RTT transcriptome varies in a context-specific manner. The authors may consider removing statements about transcriptional maturation from the manuscript and refer to these results as deficits in the expression of select sets of marker genes at different timepoints.

2. Based on the data presented in this study, it is unclear about the causal relationship between impaired neuronal activity and MeCP2-associated transcriptional aberrations. Performing qPCR analysis on a small set of genes would not be sufficient to conclude on the transcriptional programs in developing neurons or about transcriptional "maturity" of *Mecp2*-null neurons. In addition, the gene expression data reported in the manuscript demonstrate relatively subtle rescue after ampakine treatment: expression levels of only a subset of genes became more similar to the expression levels found in untreated WT neurons. Thus, the authors may consider tone down their conclusions in several occasions. Increasing neuronal activity is sufficient to ameliorate select MeCP2-associated transcriptional changes, which may suggest the possibility of phenotypic benefit of artificially increasing glutaminergic activity in vivo.

3. There are a number of instances throughout the manuscript where control groups/experiments are not included/presented. For example, the analysis of the effects of ampakine on neuronal

morphology (figure 3B, are there wild-type treated and untreated controls?), the gene expression studies with ampakine (figure 3D-G, are there vehicle treated WT and null controls?), the analysis of the effects of ampakine and nifedipine in the primary neuronal culture system (figure 4I, is there a wild-type control?), and the rotarod assay (figure 5F, can the authors include WT data in the same plot? so comparisons can be made about motor coordination at each trial). Having WT/control data included in the data presentation would allow readers to appreciate the effect of ampakine treatment.

4. The authors need to include the numbers of biological (and technical if applicable) replicates for each experiment in the figure legends (e.g. for immunofluorescence, not just number of wells, but also the number of cells imaged). This information is missing for some of the experiments/figures.

5. Although the principal components analysis is shown for each set of genes in the initial gene expression studies (Figure 1), it would be informative to include each individual genes for every set in a supplemental figure, such as those shown in figure S3.

6. This study relies on dendritic morphogenesis analysis in several of their experiments. When introducing neuronal features in RTT and MeCP2 mouse models, such as the first paragraph in the introduction, intrinsic variabilities of neuronal morphology depending on cell type, age and MeCP2 mutation need to be kept in mind when interpreting dendritic morphological data.

7. Finally, given the notion of RTT being considered as a reversible condition, it would be informative if the authors can comment on a late activity-enhancing experiment in the discussion. Would it be beneficial at all or is there a time window limited to early treatment with CX546?

Referee #2 (Remarks for Author):

This manuscript describes the effect of the AMPAkin CX546 on the transcriptional profile, responsiveness to NMDA, and intracellular Cl⁻ levels in cultured cortical neurons from Mecp2 KO embryos. In addition, the authors show improvement of life span, motor ability, and spatial learning in male Mecp2 KO mice treated with CX546 during the 1st postnatal week. The manuscript is very well written, although it needs careful checking of bibliographic references, because there are incorrect citations for statements in the text). The data seem of sufficient quality to support the authors' interpretations. However, a few major issues should be addressed to fully support the authors' conclusions.

Major questions (may require additional experiments)

1. The authors should demonstrate that CX546 reached the expected target after in vivo injections for behavioral assessments (i.e. central target engagement), like they show for in vitro treatments.

2. A functional readout of the effect of the higher intracellular Cl⁻ concentration after CX546 would provide further support for the authors' model. Is such expected switch to GABA hyperpolarizing responses also normalized after in vivo CX546 injections?

3. The output of MEA recordings (population action potentials) cannot differentiate between underlying genotype differences and CX546 effects on intrinsic excitability and synaptic activity, which reduces their impact in supporting the authors' model. How are these observations reconciled with a delayed switch to GABA hyperpolarizing responses, which would contribute to neuronal depolarization due to both glutamate and GABA?

4. What was the statistical Power yielded by the sample numbers used in all the experiments (post-hoc Power analysis)? Also, the authors should explicitly state if investigators used criteria for data inclusion and exclusion, if culture dishes and mice were randomly assigned to treatments, and if investigators were blinded of the treatments and genotypes during data acquisition and analyses (see Landis et al. Nature 2012).

Minor questions (may not require additional experiments)

5. The authors need to explain their rationale for using only male mice.

6. All bar graphs should include individual data points, or be replaced by scatter plots

7. The Results section should include numerical data (means, SDs or SEMs, n, and p values); otherwise, it reads like an Abstract Discussion section.

8. The Abstract needs more specific information and less vague language.

Referee #3 (Remarks for Author):

The study by Scaramuzza et al investigates the general hypothesis that stimulating excitatory drive in the MeCP2-deficient mouse brain will be sufficient to improve certain phenotypes in MeCP2-null cultured neurons, and later in a mouse model that recapitulate impairments seen in Rett syndrome patients. For this, they employ an ampakine drug whose actions modestly enhance AMPA receptor conductance properties. The study contains a complementary set of in vitro and in vivo experiments, and in general consistent outcomes in both systems are reported. While I am generally enthusiastic about the study, there are several issues where additional information or clarification is needed.

Major Issues

1. The premise of the study is that ampakine CX546 increases AMPA receptor responsiveness. But this is not actually demonstrated in any of the data shown. This is surprising given the use of the multi-electrode array panels shown in Figure 2. The effect on NMDA induced calcium ion responsiveness at the earlier administrative time is insufficient to show this, nor does nimedipine show a direct CX546 action on AMPA receptor activity.

2. Related to this, the authors show the effect conveyed by CX546 is more pronounced when administration is done from in vitro day 3-6. However, AMPA receptor GRIA2 expression at that time window was preserved relative to WT; it only decreased at later in vitro times when CX546 had almost no effect. If AMPA receptor prevalence is not different from wild-type at DIV 3 to 7, then why would there be a dramatic difference in outcome if drug given early?

3. Figure 2 shows a number of neurophysiological differences between cultured WT and MeCP2-null neurons. But these panels only show data from later stages of culture (DIV 18-22). How did these parameters compare at the earlier culture stages when the CX546 drug produced the largest effect? This is important since the ampakine enhances active AMPA receptors, so if their activity at this stage is equivalent between WT and null then the outcome is harder to explain. This is important since the ampakine enhances active AMPA receptors, so if their activity at this stage is

equivalent between WT and null then the outcome is harder to explain. Morphologically at DIV 3 there is no difference between WT and mutant neurons - deficits only appear at DIV 8 in the same Figure.

4. The characterization of nestin immunoreactive prevalence in the cultures at different in vitro stages is confusing as it is not clear what cultures are shown in Fig S1 (the red channel is labeled MeCP2 but the staining suggests otherwise). Moreover, the real question was whether there would be a difference in nestin prevalence between wild-type and mutant cultures at these culture stages. The reason is that the authors indicate the cultures display synchronized maturation patterns, which are important for the interpretation of the results. If differences in nestin (of ki67) exist between cultures, then the synchrony aspect becomes less clear. Would the interpretations be different for Figure 2 data if a spectrum of more juvenile neurons were present in the MeCP2-null cultures at the later assay times shown in Figure 2?

Minor Issues

1. In the Abstract and Introduction, the specific brain regions displaying hypo-excitatory basal activity should be specified. This is not ubiquitous across the entire MeCP2-deficient brain.

2. In the Introduction, the frequency of Rett syndrome amongst other severe intellectual disabilities is over-stated. For example, Down's syndrome has an incidence rate of about 1 in 2,500 female births. Perhaps the authors meant Rett syndrome is the most common monogenetic cause.....

Overall, though, the study illustrates that beneficial effects arise from early ampakine administration in this Rett syndrome mouse model, and that a fairly acute window of treatment can facilitate long-lasting effects. The data also illustrate that an earlier administration yields better outcomes than later administration. These results will be of interest to the field. My primary concern is not with the observed effects, but rather with accepting the mechanism for the effects stems from robustly enhanced AMPA receptor activity based on the data presented. This may well be the case, but showing this, and the magnitude of effect and whether treatment restores mutant neurons to normal activities during the window of treatment, would add significantly to how one mechanistically interprets the outcomes.

Reply to Reviewers

We thank all reviewers for their thorough and constructive comments. Following their observations, we have extensively modified most or added new figures, and revisited the whole text. A completely revised abstract is now presenting the manuscript. We sincerely believe that by addressing the points raised by the reviewers, our revised manuscript has been significantly improved. Please, find below a point-by-point reply.

REVIEWER 1

"This study by Scaramuzza et al sets out to determine whether enhancing neuronal activity during early brain development prevents the onset of Rett syndrome-associated molecular and cellular phenotypes. Working in an in vitro system, the authors demonstrate that Mecp2-null neurons derived from neural progenitor cells from mouse embryonic cortex exhibit deficits in neuronal activity, transcription of a select set of genes, and dendritic morphology. To test whether these effects could be attenuated by pharmacologically increasing excitatory neuronal activity, they treated progenitor-derived neurons with ampakine CX546. Treating the neurons at two different times resulted in variable effects on transcription and neuronal morphology, with greater effects achieved in response to earlier treatment. The authors then extended these findings, first to a Mecp2-null primary neuronal culture system, and then in vivo with Mecp2-null mice, demonstrating that ampakine treatment resulted in increased survival, decreases in phenotypic severity score, and rescue of behavioral deficits in two different assays.

The premise of this study is quite novel. Rett syndrome (RTT) is caused by loss of MeCP2, a nuclear protein that binds methylated DNA. Loss of MeCP2 leads to widespread transcriptional changes due to a yet-to-be-defined role of MeCP2 in the regulation of gene expression. These gene expression alterations are thought to be the primary drivers of RTT pathogenesis, leading to the cellular, circuit, and behavioral abnormalities associated with RTT in adulthood. This study, on the other hand, argues that there exists an "early molecular phase" of RTT and that impaired neuronal activity in early development may drive or promote the pathogenesis of RTT, resulting in transcriptional changes that lead to other RTT-like phenotypes. In addition to its interesting hypothesis, this study is well-conducted in general. The authors employed a previously established in vitro system and validated molecular and cellular deficits shown in Mecp2-null neurons. Additionally, the authors follow two drug treatment schedules with multiple phenotypic readouts. The experiments are conducted and analyzed in a statistically rigorous manner, with appropriate numbers of biological replicates and the correct statistical tests. Several questions/weaknesses, as outlined below, if appropriately addressed, would significantly improve the study and warrant publication in EMBO Molecular Medicine."

We thank the reviewer for the constructive comments and we are pleased to hear that the rationale of our study was found novel and interesting. Further, we are really glad to know that the study was received as well-conducted and statistically rigorous.

Point by point discussion:

1. *The authors use the term "transcriptional maturity" to describe the molecular signature of Mecp2-null neurons and argue that enhancing excitatory neuronal activity improves transcriptional maturity and attenuates cellular phenotypes. Based on the gene expression studies presented in Figure 1, the authors conclude that there is a deficit in transcriptional maturation due to changes in the expression of markers of progenitor cells at earlier time points and changes in the expression of markers of post-mitotic neurons at later time points. This does not necessarily imply a deficit in transcriptional "maturation" - to claim there is a deficit in maturation it would be necessary to show that Mecp2-null cells are "immature" at later stages,*

with expression profiles resembling an earlier stage (which does not seem to be the case based on the PCA plots). However, this could reflect deficits in transcriptional "identity" at different timepoints, particularly given the recent findings that RTT transcriptome varies in a context-specific manner. The authors may consider removing statements about transcriptional maturation from the manuscript and refer to these results as deficits in the expression of select sets of marker genes at different timepoints.

We agreed with the reviewer and revised the concept of transcriptional maturity throughout the entire manuscript. As suggested by the reviewer we introduced the concept of impaired transcriptional identity to describe the defects observed in developing *Mecp2* null samples. Moreover, we added a new transcriptional analysis made on DIV 22 NPC derived neurons (the endpoint of our differentiation protocol, this assessment is part of Expanded Figure 2). These new results now highlight the persistence of the transcriptional deregulation of select sets of marker genes during all the *Mecp2* null NPCs differentiation process. This piece of evidence fits with our previously published data describing *in vitro* and *in vivo* the transcriptional defects driven by the lack of *Mecp2* (Bedogni *et al*, 2016; Cobolli Gigli *et al*, 2018) and reinforces the involvement of genes associated with neuronal activity in the genesis of defects in null samples.

- 2. Based on the data presented in this study, it is unclear about the causal relationship between impaired neuronal activity and MeCP2-associated transcriptional aberrations. Performing qPCR analysis on a small set of genes would not be sufficient to conclude on the transcriptional programs in developing neurons or about transcriptional "maturity" of Mecp2-null neurons. In addition, the gene expression data reported in the manuscript demonstrate relatively subtle rescue after ampakine treatment: expression levels of only a subset of genes became more similar to the expression levels found in untreated WT neurons. Thus, the authors may consider tone down their conclusions in several occasions. Increasing neuronal activity is sufficient to ameliorate select MeCP2-associated transcriptional changes, which may suggest the possibility of phenotypic benefit of artificially increasing glutamergic activity in vivo.*

As for point #1, we toned down the interpretation of both the basal transcriptional defects between wt and *Mecp2* null samples and the magnitude of the rescue effects.

- 3. There are a number of instances throughout the manuscript where control groups/experiments are not included/presented. For example, the analysis of the effects of ampakine on neuronal morphology (figure 3B, are there wild-type treated and untreated controls?), the gene expression studies with ampakine (figure 3D-G, are there vehicle treated WT and null controls?), the analysis of the effects of ampakine and nifedipine in the primary neuronal culture system (figure 4I, is there a wild-type control?), and the rotarod assay (figure 5F, can the authors include WT data in the same plot? so comparisons can be made about motor coordination at each trial). Having WT/control data included in the data presentation would allow readers to appreciate the effect of ampakine treatment.*

The reviewer is absolutely correct and in the revised version of the manuscript we now provide the data on wt ctrl vs. wt CX546 treated groups. Attempting to generate easy-to-read figures, we originally chose to not include these data in the manuscript. The requested control groups are now included in dedicated supplementary figures. In particular, we are now including the morphological and transcriptional analyses made on wt maturing neurons treated with CX546 in the early and late time windows (Supplementary Figure 1) and the effects of CX546 treatment on wt animals treated from P3 to P9, including both behavioral and molecular assessments (Supplementary Figure 3). The analysis of the effects of Ampakine and Nifedipine on primary neuronal culture (Figure 4I) does not include wt ctrl and wt CX546 treated samples since we specifically aimed at proving that Nifedipine blocks the capacity of CX546 to exert its positive effects.

- 4. The authors need to include the numbers of biological (and technical if applicable) replicates for each experiment in the figure legends (e.g. for immunofluorescence, not just number of wells, but also the number of cells imaged). This information is missing for some of the experiments/figures.*

We agreed with the reviewer and included the requested information (see both Results section and Figures legends).

5. *Although the principal components analysis is shown for each set of genes in the initial gene expression studies (Figure 1), it would be informative to include each individual genes for every set in a supplemental figure, such as those shown in figure S3.*

As suggested, we completely revised the original heat map by including the expression level of each individual gene at all the selected time points (DIV3, DIV8, DIV14; Expanded Figure 2). As in the previous version of the heatmap, genes are clustered depending on their functions.

6. *This study relies on dendritic morphogenesis analysis in several of their experiments. When introducing neuronal features in RTT and MeCP2 mouse models, such as the first paragraph in the introduction, intrinsic variabilities of neuronal morphology depending on cell type, age and MeCP2 mutation need to be kept in mind when interpreting dendritic morphological data.*

We agreed with the reviewer's comment and we regret having been rather vague on this point. We added a new sentence regarding this topic in the first paragraph of the Introduction: "*Defective neuronal features have been reported as well, including reduced soma size, dendritic branching, number of spines and synaptic contacts (Guy et al, 2001; Bedogni et al, 2016; Baj et al, 2014; Belichenko et al, 2009; Chao et al, 2007; Fukuda et al, 2005; Kishi and Macklis, 2004; Sampathkumare et al, 2016; Rietveld et al, 2015). Importantly, although these phenotypes vary depending on the cell type, the age and the type of Mecp2 mutations, soma size is consistently reduced throughout development of RTT syndrome mouse models, therefore appearing as a robust and reliable biomarker (Wang et al, 2013)*".

7. *Finally, given the notion of RTT being considered as a reversible condition, it would be informative if the authors can comment on a late activity-enhancing experiment in the discussion. Would it be beneficial at all or is there a time window limited to early treatment with CX546?*

A new comment concerning this point was added in the discussion, where we propose to focus on different time windows for CX546 administration, including pre- and symptomatic stages. This can be found at the end of Discussion. The enhancement of *Mecp2* null neuronal activity has already been proposed in other important studies as a tool to ameliorate defects even in adult ko mice. However, these studies did not analyze the effects of the drug on animal behavior and did not investigate persistence of positive effects. Our *in vitro* studies suggest that early interventions might be more effective. In the future we will investigate whether a later administration of the drug to *Mecp2* null mice weakens the positive effects therefore highlighting the existence of a most efficacious time window of treatment. Further, as now stated in the discussion, we will investigate the advantages of intermitted and repetitive treatments.

REVIEWER 2

This manuscript describes the effect of the AMPAkinic CX546 on the transcriptional profile, responsiveness to NMDA, and intracellular Cl⁻ levels in cultured cortical neurons from Mecp2 KO embryos. In addition, the authors show improvement of life span, motor ability, and spatial learning in male Mecp2 KO mice treated with CX546 during the 1st postnatal week. The manuscript is very well written, although it needs careful checking of bibliographic references, because there are incorrect citations for statements in the text). The data seem of sufficient quality to support the authors' interpretations. However, a few major issues should be addressed to fully support the authors' conclusions.

We thank the reviewer for the positive comments. We apologize for incorrect citations; we have carefully revised the whole manuscript and we hope to have been able to solve any problem.

Point by point discussion:

1. *The authors should demonstrate that CX546 reached the expected target after in vivo injections for behavioral assessments (i.e. central target engagement), like they show for in vitro treatments.*

It was already proved that Ampakines are able to increase in diseased brains, including the *Mecp2* null brain, *Bdnf* expression (Bretin *et al*, 2017; Ogier *et al*, 2007). We have thus decided to use *Bdnf* mRNA levels as readout of the capacity of CX546 to reach the mouse RTT brain. To this purpose, as depicted in Figure 5I, we assessed *Bdnf* transcriptional levels in the cerebral cortex of animals treated with CX546 (or vehicle) following the same timing depicted in Figure 5, panel A. We demonstrate that the reduced expression of *Bdnf* in *Mecp2* null cortices is rescued after CX546 treatment (Panel I). Moreover, we detected a trend towards increased levels of *Bdnf* after CX546 treatment also in wt samples (Supplementary Figure 3F, $p=0,07$). This is another evidence that reinforces the fact that CX546 reaches the desired targets *in vivo*. These data are reinforced by the ability of CX546 to rescue, in the *Mecp2* null cortex, *Kcc2* transcription (Figure 5, panel J). We believe the addition of these evidence has truly strengthened our analysis; we thank the reviewer for suggesting these experiments.

2. *A functional readout of the effect of the higher intracellular Cl^- concentration after CX546 would provide further support for the authors' model. Is such expected switch to GABA hyperpolarizing responses also normalized after in vivo CX546 injections?*

As mentioned above, in line with the rescued levels of *Bdnf* expression (Figure 5I), we have added new evidence about GABA signaling normalization after CX546 treatment *in vivo*. In fact, Figure 5 panel J shows that the expression of *Kcc2* is rescued in cortical samples of P45 animals treated with Ampakine from P3 to P9. This evidence fits with the *in vitro* data depicted in Figure 4. As for point #1, we thank the reviewer for suggesting these new assessments.

3. *The output of MEA recordings (population action potentials) cannot differentiate between underlying genotype differences and CX546 effects on intrinsic excitability and synaptic activity, which reduces their impact in supporting the authors' model. How are these observations reconciled with a delayed switch to GABA hyperpolarizing responses, which would contribute to neuronal depolarization due to both glutamate and GABA?*

The Reviewer is right in saying that the network activity results from both intrinsic excitability and balance between excitatory and inhibitory synaptic transmission. We used MEA recordings as an important proxy of network maturation that consists of increased expression of ion channels, as well as of the assembly of a progressively increasing network of synaptic connections. In our experiments, the mutant networks had a clearly immature activity that increased very little, compared to wild type networks. Regarding the delayed switch to GABA hyperpolarizing responses, we agree with the Reviewer that MEA recordings are not the best method to detect them. Indeed, in addition to intrinsic excitability, the overall MEA activity is more dependent on changes of synaptic strength due to short-term plasticity (i.e. facilitation/depression) than on the excitation/inhibition balance of basal synaptic strength (e.g., in response to single pulses). Thus, we do not see any discrepancy between the observed delay in the GABA switch and the slowed increase of spontaneous firing in mutant networks. Rather they are likely two aspects of the same underlying developmental defect.

4. *What was the statistical Power yielded by the sample numbers used in all the experiments (post-hoc Power analysis)? Also, the authors should explicitly state if investigators used criteria for data inclusion and exclusion, if culture dishes and mice were randomly assigned to treatments, and if investigators were blinded of the treatments and genotypes during data acquisition and analyses (see Landis et al. Nature 2012).*

We are sincerely grateful to the reviewer for raising the point and citing the important study by Landis *et al.* In Statistical analysis we now write that *“each culture wells from in vitro analysis and mice from in vivo experiments were randomly assigned to treatments, and for all the experiments reported in the manuscript the investigators were blinded to the treatments and genotypes during data acquisition and analyses.”* In spite of this novel statement, in Materials and Methods we have maintained previous texts defining when the operator was blind to the genotype and treatments (see Morphological analyses, Neurobehavioral characterization and behavioral assessment). Further, as made explicit in Statistical analysis, only samples that were identified with Grubb’s test as possible outliers were discarded from analyses. Concerning the statistical Power, we believe that the Reviewer is asking us how sample size calculation was obtained. We based sizing for *in vivo* assessments on our previous experience working on *Mecp2* null mice, so we set magnitude of variation to 35% vs. controls and a standard deviation of 10% (G*Power, 2way ANOVA repeated measures, alpha: 0.05, power 0.8). This “basic” sizing was thoroughly used to plan the experiments described in this study and as a guideline to draw our animal protocol. However, we adapted sizing to our need in many cases, as post-hoc statistic was eventually used to address the strength of our analyses.

Minor questions:

5. *The authors need to explain their rationale for using only male mice.*

We added a new statement in in the results section explaining the reason for using only male mice. The new statement is *“Although RTT mainly affects females, explorative studies are generally performed on mutant male mice that present more robust and consistent phenotypes”*.

6. *All bar graphs should include individual data points, or be replaced by scatter plots.*

As suggested by the Reviewer, we edited all graphs to include individual data points.

7. *The Results section should include numerical data (means, SDs or SEMs, n, and p values); otherwise, it reads like an Abstract Discussion section.*

We revised the Results section to include numerical data as suggested. We thank the reviewer, as the section now reads much more detailed.

8. *The Abstract needs more specific information and less vague language.*

We agreed with the Reviewer. The new manuscript now includes a deeply revised Abstract that contains more information and we believe appears less vague. For convenience we add here the new text. *“MECP2 mutations cause Rett syndrome (RTT), a severe and progressive neurodevelopmental disorder mainly affecting females. Although RTT patients exhibit a delayed onset of symptoms, several evidences demonstrate that MeCP2 deficiency alters early development of the brain. Indeed, during early maturation, Mecp2 null cortical neurons display widespread transcriptional changes, reduced activity and defective morphology. In physiological conditions these three elements are linked in a feed-forward cycle where neuronal activity drives transcriptional and morphological changes that further increase network maturity. We hypothesized that the enhancement of neuronal activity during early neuronal maturation might prevent the onset of RTT-typical molecular and cellular phenotypes. Accordingly, we show that the enhancement of excitability, obtained adding to neuronal cultures Ampakine CX546, rescues transcription of several genes, neuronal morphology and responsiveness to stimuli. Greater effects are achieved in response to earlier treatments. In vivo, short and early administration of CX546 to Mecp2 null mice prolongs lifespan, delays the disease progression and rescues motor abilities and spatial memory, thus confirming the value for RTT of an early restoration of neuronal activity.”*

REVIEWER 3

The study by Scaramuzza et al investigates the general hypothesis that stimulating excitatory drive in the MeCP2-deficient mouse brain will be sufficient to improve certain phenotypes in MeCP2-null cultured neurons, and later in a mouse model that recapitulate impairments seen in Rett syndrome patients. For this, the employ an ampakine drug whose actions modestly enhance AMPA receptor conductance properties. The study contains a complementary set of in vitro and in vivo experiments, and in general consistent outcomes in both systems are reported. While I am generally enthusiastic about the study, there are several issues where additional information or clarification is needed.

We really thank the Reviewer for the encouraging and positive comments and for the enthusiasm reported for our work.

Point by point discussion:

Major Issues

1. *The premise of the study is that ampakine CX546 increases AMPA receptor responsiveness. But this is not actually demonstrated in any of the data shown. This is surprising given the use of the multi-electrode array panels shown in Figure 2. The effect on NMDA induced calcium ion responsiveness at the earlier administrative time is insufficient to show this, nor does nimedipine show a direct CX546 action on AMPA receptor activity.*

In order to respond to the reviewer’s comment and to verify the involvement of AMPA receptor in the effects driven by CX546, we have added a novel panel (Expanded Figure 3B) in which we show that the simultaneous addition of CX546 and

NBQX (a competitive AMPA receptors antagonist; Twelve *et al*, 2015; Chen *et al*, 2017) results in a significantly reduced ability of CX546 to induce the expected intracellular responses assessed through the phosphorylation of AKT. This is in line with published results on the pharmacology of CX546 (Lynch and Gall, 2006). In the same study Lynch and Gall described CX546 as an “indirect agonist” of the AMPA receptor, as it facilitates AMPA functions selectively when the receptor is bound by glutamate. This implies that the exposure to CX546 enhances the ability of neurons to respond to stimuli, which results in the overall enhancement of network functions.

2. *Related to this, the authors show the effect conveyed by CX546 is more pronounced when administration is done from in vitro day 3-6. However, AMPA receptor GRIA2 expression at that time window was preserved relative to WT; it only decreased at later in vitro times when CX546 had almost no effect. If AMPA receptor prevalence is not different from wild-type at DIV 3 to 7, then why would there be a dramatic difference in outcome if drug given early?*

The reviewer is right in saying that at initial time points there is no difference in the expression of the Gria2 subunit. However, the use of CX546 was not intended to link the defects displayed by *Mecp2* null samples with defective AMPA receptor functions but, rather, to assess the link between poor maturity of null neurons and poor network functions in general. As a matter of fact, by measuring Gria2 expression at early time points we assessed whether *Mecp2* null neurons were biochemically competent to respond to the drug. The main finding of this experiment is not the recovery of AMPA signaling but the ability of globally enhanced network excitability to prevent the establishment of the typically RTT neuronal phenotypes later in development. Indeed, it should be considered that CX546 is able to trigger the engagement of many activity-dependent players (e.g. voltage gated channels, calcium-binding proteins, kinases, second messengers), which may act synergistically to boost neuronal maturation. This conclusion is further reinforced by evidence obtained with the new experiments described in Expanded Figure 4. By inducing a broad depolarization through the exposure of cell cultures to 4 mM KCl, we rescued typical morphological defects of *Mecp2* null neurons. To be noticed, this evidence mimics the results obtained with CX546.

3. *Figure 2 shows a number of neurophysiological differences between cultured WT and MeCP2-null neurons. But these panels only show data from later stages of culture (DIV 18-22). How did these parameters compare at the earlier culture stages when the CX546 drug produced the largest effect? This is important since the ampakine enhances active AMPA receptors, so if their activity at this stage is equivalent between WT and null then the outcome is harder to explain. Morphologically at DIV 3 there is no difference between WT and mutant neurons - deficits only appear at DIV 8 in the same Figure.*

To assess whether neuronal activity was impaired in the very early stages of NPCs maturation we couldn't use MEA approach, since this technique allows detection of electrical signal only when synaptic connections are already established. To answer the reviewer's question we thus performed calcium imaging. As displayed in the graph here reported, we highlighted no difference between wt and null NPCs in calcium transient after KCl exposure. This is not surprising, given the absence of morphological defects at such early stage that the reviewer pointed out. However, in this *in vitro* system, cells at DIV3 are very immature, which obviously makes highlighting differences between groups particularly difficult. In fact, at this stage both

Nestin positive and Tuj1 positive cells are frequently found in both wt and null cultures (Expanded Figure 1). Since it is technically difficult to distinguish between these two types of cells based on morphology, we are not willing to include this study in the final manuscript. However, our PCA analyses (Figure 1) suggest that wt and null neurons start to exhibit a different expression of post-mitotic markers already at DIV3. Further, we show that *Mecp2* null primary neurons display altered intracellular chloride level already starting from DIV3 (Figure 4 panel). Accordingly, we already demonstrated that null cortical primary neurons display reduced calcium transients after chemical and electrical stimulation as early as at DIV3 (Bedogni *et al*, 2016). In accordance with Spitzer publication (Spitzer, 2006), we believe that the observed small transcriptional defects affecting null NPCs during early differentiation contribute to the later reduced capacity of *Mecp2* null neurons to respond to stimuli. For this reason, we applied our strategy of early activity enhancement. In accordance with the hypothesis, our data show that the early Cx546 treatment prevents the worsening of transcriptional defects and the consequent establishment of functional and morphological defects. This is also strengthened by the study described in Expanded Figure 4, where we show that the exposure to KCl from DIV0 to DIV6 has more potential in rescuing morphological defects compared to the exposure from DIV7 to DIV14 (panel E-G). All in all, these observations really stress the highly dynamic nature of the processes that enable neuronal maturation and the relevance of neuronal activity for neuronal maturation and functioning.

4. *The characterization of nestin immunoreactive prevalence in the cultures at different in vitro stages is confusing as it is not clear what cultures are shown in Fig S1 (the red channel is labeled MeCP2 but the staining suggests otherwise). Moreover, the real question was whether there would be a difference in nestin prevalence between wild-type and mutant cultures at these culture stages. The reason is that the authors indicate the cultures display synchronized maturation patterns, which are important for the interpretation of the results. If differences in nestin (of ki67) exist between cultures, then the synchrony aspect becomes less clear. Would the interpretations be different for Figure 2 data if a spectrum of more juvenile neurons were present in the MeCP2-null cultures at the later assay times shown in Figure 2?*

As suggested by the reviewer we assessed if there was a difference in the presence of Nestin positive cells between cultures. As depicted in the graph reported in the Expanded Figure 1C we reported equal number of Nestin positive cells between wt and *Mecp2* null NPC at DIV8. Further, the levels of *Ki67* expression are not different between wt and null DIV0 cultures (Expanded Figure 2A). This piece of evidence is in line with previous published results (Cobolli Gigli *et al*, 2018), where we showed both *in vitro* (using the same model) and *in vivo* no gross differences in the dynamics of cell cycle exit between genotypes, but, rather, differences in the transcriptional identity of progenitors, which is confirmed here by our PCA plots (Figure 1P,p'). In wt cultures we detected Nestin positive cells (in green) and Map2 positive cells (in red) in culture at DIV8 (Expanded Figure 1A,B); as expected, the number of Nestin positive cells strongly diminished at DIV22.

Minor issues

5. *In the Abstract and Introduction, the specific brain regions displaying hypo-excitatory basal activity should be specified. This is not ubiquitous across the entire MeCP2-deficient brain.*

As suggested by the reviewer, we rephrased the description of the neuronal activity dysfunction in different brain regions. The new paragraph dedicated to this topic is in the Introduction: "Besides morphological alterations, impaired neuronal functions were also observed in adult mice, resulting in a complex derangement of brain activity (Nelson & Valakh, 2015). *Mecp2* null cortical neurons feature reduced activity caused by both a selective impairment in excitatory transmission and a reduced connectivity between excitatory neurons (Dani *et al* 2005; Dani *et al.*, 2009; Shepherd and Katz 2011; Sceniak *et al.*, 2016). *Mecp2* KO visual cortex manifest similar deficiency in neuronal and network activity; this, however, appears to result from stronger inhibition (Durand *et al.*, 2012). On the contrary, the adult and symptomatic *Mecp2* null hippocampus

(but not the pre-symptomatic one) suffers from elevated neuronal activity and occluded LTP caused by potentiated synapses (Li *et al.*, 2017)".

6. *In the Introduction, the frequency of Rett syndrome amongst other severe intellectual disabilities is over-stated. For example, Down's syndrome has an incidence rate of about 1 in 2,500 female births. Perhaps the authors meant Rett syndrome is the most common monogenetic cause.*

The reviewer is right. Down's syndrome is the first cause of intellectual disability worldwide; however, it is generally not considered a severe condition. On the contrary Rett syndrome is very severe and because of its incidence is thus considered the first cause of *severe* intellectual disability in girls.

24th Dec 2020

Dear Prof. Landsberger,

Thank you for the submission of your revised manuscript to EMBO Molecular Medicine. I would also like to thank you for your assistance in clarifying the issue with figure aberration and for replacing the mosaic image assembled for Fig 1B with an unmodified excerpt of the native source data image. I am pleased to inform you that we will be able to accept your manuscript pending the following final amendments:

- 1) With approaching holidays and the end of the year we encountered high number of submissions, so that our data editors were not able to process all received manuscripts before the holiday season. Therefore, we will send you the document with data editor's suggestions after the holidays and as soon as our data editors process your manuscript. Please do not submit your revised manuscript before we send you the file with data editor's suggestions. Thank you for your understanding.
- 2) Please address all the referee's concerns. Further experiments to address referee #2 concerns would be appreciated but not required. However, all the referee's concerns should be addressed at least in writing.

***** Reviewer's comments *****

Referee #1 (Remarks for Author):

In this revised manuscript, the authors have conducted additional experiments, re-analyzed some of the data, included details on biological replicates, and modified numerous conclusions in response to critiques raised in the first round of review. The manuscript is significantly improved and would meet the merit of acceptance in EMBO Molecular medicine. A few minor points on wording/data presentation are suggested for the authors to consider.

1. The authors describe in the Abstract: "In physiological conditions these three elements are linked in a feed-forward cycle where neuronal activity drives transcriptional and morphological changes that further increase network maturity." This sentence states that this is an established phenomenon. To the best of my knowledge, this is a hypothesis about brain development. Of course, in RTT, it is unknown whether this occurs to greater extent.
2. In the first paragraph of the introduction: The sentence about hemizygous *Mecp2* null male mice sounds overly stated. There are very few male RTT patients as most may not make it to birth.
3. The authors have toned down numerous statements in the abstract about transcriptional maturity, but in the main text "transcriptional maturity" is continuously referred. For instance, in the introduction the authors repeatedly refer to transcriptional maturity. The argument would be better characterized as: reduced neuronal activity in early development leads to impaired transcriptional identity (see my comment in the first round of review to avoid potential confusion to readers).
4. The authors present PCA plots in figures 1 and 3 to illustrate differences in transcriptional identity. However, the differences between biological replicates within each genotype appear to be larger than the differences between the two genotypes. Thus, honing in on small distances in multidimensional space based off of the expression of a small number of genes isn't that convincing whether or not there is truly a difference in transcriptional identity. A heatmap, or pointing out the specific DEGs, might be more effective to make the point that the transcriptional identities are impaired at different developmental timepoints (and can be rescued by increasing neuronal activity). The authors may show the PCA plots in supplemental figures if applicable.

Referee #2 (Comments on Novelty/Model System for Author):

The medical impact is low due to the lack of data on the effects of the ampakine in female *Mecp2* Het mice. The majority of studies testing a novel therapeutic strategy for RTT (like ampakines to increase BDNF levels) are done in female *Mecp2* Het mice to account for the contribution of a mosaic expression pattern of the mutant allele.

Referee #2 (Remarks for Author):

EMM-2020-12433V2

Scaramuzza et al.

This manuscript describes the effect of the AMPAkinase CX546 on the transcriptional profile, responsiveness to NMDA, and intracellular Cl⁻ levels in cultured cortical neurons from Mecp2 KO embryos. In addition, the authors show improvement of life span, motor ability, and spatial learning in male Mecp2 KO mice treated with CX546 during the 1st postnatal week. The manuscript is very well written and the data seem of sufficient quality to support the authors' interpretations. The responses to the previous review are satisfactory in general, but some issues still remain.

Major questions (may require additional experiments)

1. Regarding a functional readout, have the authors tested if Mecp2 KO neurons treated with CX546 show the expected consequence of higher intracellular Cl⁻ concentration, i.e. hyperpolarizing responses to GABA (resulting from higher Kcc2 expression)?
2. Regarding the contribution of GABAergic inhibition to "network maturation" of cultured neurons on MEAs, have the authors tested if a GABA-A antagonist (e.g. picrotoxin) has a smaller effect on spontaneous firing (from disinhibition of the majority of neurons on the MEA) in Mecp2 KO neurons (because GABA should still be depolarizing)? Also, have they tested if this effect of disinhibition is normalized in Mecp2 KO neurons by CX546? In other words, a confirmation that CX546 led to the proper maturation of GABAergic inhibition in Mecp2 KO neurons.
3. It is true that most studies of Mecp2 function and of the consequences of its deletion are done in male Mecp2 KO mice for the reasons given, but the majority of studies testing a novel therapeutic strategy (like ampakines to increase BDNF levels) are done in female Mecp2 HET mice to account for the contribution of a mosaic expression pattern of the mutant allele. Have the authors performed exploratory/pilot studies of the effect of CX546 in female Mecp2 HET mice?

EMM-2020-12433V2 Scaramuzza et al: The enhancement of activity rescues the establishment of *Mecp2* null neuronal phenotypes**Point-by-point comment to the reviewers' notes**

Referee #1 (Remarks for Author): In this revised manuscript, the authors have conducted additional experiments, re-analyzed some of the data, included details on biological replicates, and modified numerous conclusions in response to critiques raised in the first round of review. The manuscript is significantly improved and would meet the merit of acceptance in EMBO Molecular medicine. A few minor points on wording/data presentation are suggested for the authors to consider.

1. The authors describe in the Abstract: "In physiological conditions these three elements are linked in a feed-forward cycle where neuronal activity drives transcriptional and morphological changes that further increase network maturity." This sentence states that this is an established phenomenon. To the best of my knowledge, this is a hypothesis about brain development. Of course, in RTT, it is unknown whether this occurs to greater extent.

The reviewer observation is correct, and we modified the text in order to make it clear that the existence of this feed-forward cycle is in fact still a hypothesis. By slightly modifying the abstract, the introduction and the discussion we made this very clear.

2. In the first paragraph of the introduction: The sentence about hemizygous *Mecp2* null male mice sounds overly stated. There are very few male RTT patients as most may not make it to birth.

*We agreed and modified the text making it clear that the highly reduced life span featured by male *Mecp2* KO mice is not typical of RTT female patients. We decided to not discuss whether several male RTT patients do not make it to birth because this topic, which in any case is not relevant for the manuscript, would deserve a long discussion. Indeed, since *MECP2* mutations mainly occur in sperm, they are generally transmitted to females. Further, in rodents there is no evidence of deviation from Mendel inheritance therefore making this issue quite complicate.*

3. The authors have toned down numerous statements in the abstract about transcriptional maturity, but in the main text "transcriptional maturity" is continuously referred. For instance, in the introduction the authors repeatedly refer to transcriptional maturity. The argument would be better characterized as: reduced neuronal activity in early development leads to impaired transcriptional identity (see my comment in the first round of review to avoid potential confusion to readers).

We carefully read the manuscript to amend any reference to transcriptional maturity; as suggested, we now only refer to the concept of morphological and functional immaturity.

4. The authors present PCA plots in figures 1 and 3 to illustrate differences in transcriptional identity. However, the differences between biological replicates within each genotype appear to be larger than the differences between the two genotypes. Thus, honing in on small distances in multidimensional space based off of the expression of a small number of genes isn't that convincing whether or not there is truly a difference in transcriptional identity. A heatmap, or pointing out the specific DEGs, might be more effective to make the point that the transcriptional identities are impaired at different developmental timepoints (and can be rescued by increasing neuronal activity). The authors may show the PCA plots in supplemental figures if applicable.

We agreed with this comment and modified figure 1. In particular, as requested, we have replaced the panels depicting PCA plots with a new panel showing the transcriptional levels of selected DEGs. PCA panels are now part of the revised Expanded Figure 2 and replace the previously proposed heat map (previous Figure EV2). Concerning Figure 3, PCA plots (panel D-F) have been maintained but the text has been modified to make it clear that these are qualitative visualizations of CX546 effects. These plots corroborate the representation of DEGs in the different experimental groups (panels G-H).

Referee #2 (Comments on Novelty/Model System for Author): The medical impact is low due to the lack of data on the effects of the ampakine in female *Mecp2* Het mice. The majority of studies testing a novel therapeutic strategy for RTT (like ampakines to increase BDNF levels) are done in female *Mecp2* Het mice to account for the contribution of a mosaic expression pattern of the mutant allele.

Referee #2 (Remarks for Author): EMM-2020-12433V2 Scaramuzza et al. This manuscript describes the effect of the AMPAkinase CX546 on the transcriptional profile, responsiveness to NMDA, and intracellular Cl^- levels in cultured cortical neurons from *Mecp2* KO embryos. In addition, the authors show improvement of life span, motor ability, and spatial learning in male *Mecp2* KO mice treated with CX546 during the 1st postnatal week. The manuscript is very well written and the data seem of sufficient quality to support the authors' interpretations. The responses to the previous review are satisfactory in general, but some issues still remain.

Major questions (may require additional experiments)

As a general remark to the reviewer's comments, all the experiments suggested as further revision are indeed interesting, but also quite difficult to accomplish and time consuming, even more so in RTT models. In fact, it is well known that MeCP2 deficiency contextually affects several systems, sometimes with opposite molecular and functional readouts, which makes it very difficult to propose a single mechanism to explain such a heterogeneous range of phenotypes.

1. Regarding a functional readout, have the authors tested if *Mecp2* KO neurons treated with CX546 show the expected consequence of higher intracellular Cl^- concentration, i.e. hyperpolarizing responses to GABA (resulting from higher *Kcc2* expression)?

*The higher *Kcc2* expression is certainly very interesting and deserves further studies. However, it does not appear to be the only cause of the impaired morphological and functional maturation of MeCP2-deficient networks, which results from a number of wide spread transcriptional deregulations and complex impairments in network development. Moreover, a change in the intracellular chloride concentration not only affects the polarity of GABAergic transmission, but many other conductances and transporters that may contribute to the *Mecp2* developmental phenotype. Because of that we did not include any of these studies in this manuscript.*

2. Regarding the contribution of GABAergic inhibition to "network maturation" of cultured neurons on MEAs, have the authors tested if a GABA-A antagonist (e.g. picrotoxin) has a smaller effect on spontaneous firing (from disinhibition of the majority of neurons on the MEA) in *Mecp2* KO neurons (because GABA should still be depolarizing)? Also, have they tested if this effect of disinhibition is normalized in *Mecp2* KO neurons by CX546? In other words, a confirmation that CX546 led to the proper maturation of GABAergic inhibition in *Mecp2* KO neurons.

*The network activity in MEAs, depends on both intrinsic excitability and synaptic transmission and is fundamentally driven by excitatory transmission. We did not test GABA_A receptor antagonists in our MEA experiments, since they MEA studies were meant to monitor the progressive development and maturation of the network firing activity over time in culture. Moreover, being ion channels and synaptic connections globally altered in *Mecp2* KO networks, the outcome of the blockade of GABAergic transmission cannot be easily interpreted.*

Regarding the increase in intracellular chloride, it does not occur in all neurons and was measured in early stages of development (up to 9 DIV), whereas the emergence of significant network activity starts to be detectable only in later stages (14-21 DIV). This makes unlikely to detect a clear effect of a GABA shift that is not shared by the entire neuronal population.

*Importantly, the similar fold change in firing rate of WT and *Mecp2* KO networks in response to 4AP (a convulsant drug that concomitantly increases the intrinsic excitability of both glutamatergic and GABAergic neurons) argues against a widespread effect of excitatory GABA that would have changed the excitatory/inhibitory ratio in *Mecp2* KO neurons with respect to WT networks in response to 4AP.*

3. It is true that most studies of Mecp2 function and of the consequences of its deletion are done in male Mecp2 KO mice for the reasons given, but the majority of studies testing a novel therapeutic strategy (like ampakines to increase BDNF levels) are done in female Mecp2 HET mice to account for the contribution of a mosaic expression pattern of the mutant allele. Have the authors performed exploratory/pilot studies of the effect of CX546 in female Mecp2 HET mice?

The reviewer is right and we will soon start a thorough CX456 pre-clinical study aimed at defining the best timing of treatment and the effects on females. A very short sentence highlighting the importance of performing studies on females has been now added in the discussion.

The authors performed the requested editorial changes.

27th Jan 2021

Dear Prof. Landsberger,

We are pleased to inform you that your manuscript is accepted for publication.

Corresponding Author Name: Nicoletta Landsberger, Ph.D, Francesco Bedogni, Ph.D.

Manuscript Number: EMM202012433